# Fast, Secure, and High-Capacity Image Watermarking with Autoencoded Text Vectors

## Abstract

Most image watermarking systems focus on robustness, capacity, and imperceptibility while treating the embedded payload as meaningless bits. This bit-centric view imposes a hard ceiling on capacity and prevents watermarks from carrying useful information. We propose LatentSeal, which reframes watermarking as semantic communication: a lightweight text autoencoder maps full-sentence messages into a compact 256-dimensional unit-norm latent vector, which is robustly embedded by a finetuned watermark model and secured through a secret, invertible rotation. The resulting system hides full-sentence messages, decodes in real time, and survives valuemetric and geometric attacks. It surpasses prior state of the art in BLEU-4 and Exact Match on several benchmarks, while breaking through the long-standing 256-bit empirical payload ceiling. It also introduces a statistically calibrated score that yields a ROC AUC score of 0.97-0.99, and practical operating points for deployment. By shifting from bit payloads to semantic latent vectors, LatentSeal enables watermarking that is not only robust and high-capacity, but also secure and interpretable, providing a concrete path toward provenance, tamper explanation, and trustworthy AI governance. Models, training and inference code, and data splits will be available upon publication (code available in the supplementary zip file).

## 1 Introduction

Image watermarking primarily tackles authenticity verification, copy protection, or tampering localization. Recent regulatory developments, such as the EU AI Act and Chinese AI Governance rules, place watermarking among the most practical technical tools for ensuring trustworthy AI content detection. Watermarking schemes broadly fall into two categories: zero-bit watermarking, which merely indicates the presence or absence of a watermark, and multi-bit watermarking, which allows for the decoding of a hidden message.

In robust watermarking schemes, the primary objective is to ensure the watermark's survival against diverse attacks, including compression, filtering, noise, scaling, rotation, and cropping. Achieving high robustness often requires embedding the watermark redundantly or in perceptually significant (yet imperceptible) image regions, which inherently limits effective capacity. A fundamental trade-off exists: increasing capacity often compromises imperceptibility (leading to larger image distortions), while enhancing robustness typically requires redundancy, reducing capacity or secrecy.

Previous work achieved at most 100 bits of capacity (Bui et al., 2023), which is not enough in our setting. The most advanced robust watermarking schemes, leveraging deep learning, recently achieved capacities of a few hundred bits (e.g., 256 bits for VideoSeal (Fernandez et al., 2024)). While impressive in terms of robustness and imperceptibility, this bit-centric framing leaves much of the channel underutilized. The embedded message is usually a meaningless identifier, and existing approaches rarely provide any security mechanism.

**We argue for a paradigm shift: watermarking as semantic communication.** Instead of embedding arbitrary bit sequences, we embed *meaningful textual messages* represented as continuous latent vectors. Concretely, we train a lightweight text autoencoder to map sentences into a 256-dimensional latent space, which is then embedded by a finetuned watermarking model and secured through an invertible, key-conditioned rotation. At extraction, the latent vector is inverted and decoded back into natural language, producing a watermark that is robust, secure, and interpretable. At

inference time, those sentences may be sourced from human operators aiming to transmit a specific message or from an image captioning model, such as BLIP2 (Li et al., 2023) or Florence (Xiao et al., 2023).

Within this framework, watermarking is no longer a raw bit pipe but a communication channel. A compelling application is image tampering detection: by embedding a textual description of the original image, our method allows not only for the detection of manipulation but also for identifying the nature of the modification. Unlike semi-fragile or localized watermarking (Sander et al., 2025; Hu et al., 2025), which primarily indicate *if* and *where* an image has been tampered with, semantic watermarking additionally reveals *what* has been altered, as illustrated in Fig. 1.

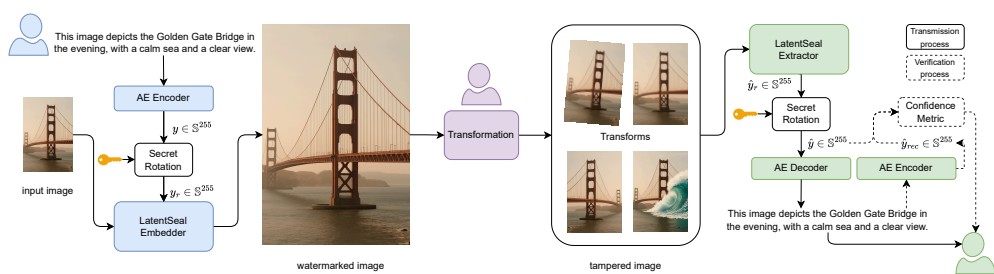

Figure 1: LatentSeal overview illustrated through an image tampering detection application. Bob writes a message $m$ describing the image's content, then feeds it to the autoencoder which outputs a latent vector $y$. To introduce security, we apply a secret rotation to $y$ conditioned by a secret key and obtain $y_r$. LatentSeal embeds $y_r$ within the input image. During transmission, the image is intercepted and transformed by Eve. Finally, Alice receives the image, performs the watermark extraction process, and recovers an estimated rotated latent vector $\hat{y}_r$. Using her secret key, identical to Bob's, she applies the inverse rotation to $\hat{y}_r$ and decodes the resulting latent vector $\hat{y}$ to reconstruct the original textual message. By comparing the received image's content with the decoded message $\hat{m}_r$, Alice can detect discrepancies and flag tampering accordingly.

The core challenge lies in embedding substantial textual information as an imperceptible watermark within a limited payload. Evennou et al. (2024) explored this direction using the lossless text compressor LLMZip (Valmeekam et al., 2023), but their payload was limited to 72 bits. Our work overcomes this bottleneck by introducing a robust text autoencoder tailored for watermarking. Adapting VideoSeal (Fernandez et al., 2024), we train a system to embed 256-dimensional unit-norm vectors, which is aligned with the latent space of the text autoencoder.

Our system operates as follows: the autoencoder encodes a message into a latent vector that is embedded into the image. Upon extraction, the latent is decoded back into the original message. Crucially, working in this latent space enables a security layer: an invertible, key-parameterized rotation ensures that only authorized decoders can recover the correct representation. Moreover, the structure of the latent space allows us to provide a confidence metric that detects unreliable extractions, enhancing trustworthiness at inference time.

Overall, our latent vector watermarking framework, named LatentSeal, significantly advances the state of the art by uniting robustness, capacity, security, and semantic utility. Compared to the strongest baselines (VideoSeal + LLMZip), LatentSeal achieves higher BLEU-4 and Exact Match, breaks through the 256-bit payload ceiling, and introduces practical mechanisms for security and reliability. Experiments on diverse datasets including captions and Wikipedia-like text demonstrate its performance, and we additionally verify that our dataset splits avoid memorization effects.

In summary, our contributions are:

- **LatentSeal:** a new, fast, and secure framework for watermarking natural language in images, including a confidence metric on decoded text that enables decision-making about authenticity.

- **Robust text autoencoder:** a lightweight autoencoder that maps sentences into a 256-D latent space specifically optimized for transmission through the watermark channel.

- **Robust and secure embedding:** we adapt VideoSeal to embed 256-D unit-norm vectors into images, and augment it with a plug-in latent-space encryption layer based on Haar-random rotations from SO(256) keyed by a secret seed. This combination yields both state-of-the-art robustness and an effective cryptographic safeguard at negligible cost.
- **Confidence scoring:** a well-grounded mechanism to flag unreliable extractions, increasing trustworthiness in deployment scenarios.
- **Extensive evaluation:** experiments on COCO-2017, PixMo-Cap, and WikiText-103 highlighting new capabilities for watermarking textual information, surpassing prior work in robustness, capacity, speed, and reconstruction quality.

By reframing watermarking from bit payloads to semantic latent vectors, LatentSeal opens a new research direction where watermarks are not only robust and imperceptible but also secure, interpretable, and directly useful for provenance and tamper explanation.

## 2 RELATED WORK

**Image Watermarking** End-to-end deep watermarking treats the task as learning an embedder–extractor pair that hides an $L$-bit string in a cover image and recovers it after attacks. Pioneering CNN systems like HiDDeN (Zhu et al., 2018) introduced differentiable distortion layers to gain robustness to JPEG, cropping and noise; later self-supervised variants (e.g. SSL-Watermark (Fernandez et al., 2022)) removed the need for labeled data. Capacity and fidelity were boosted by TrustMark (Bui et al., 2023), which couples a UNet encoder with learned error-correcting codes to reach up to 100 bits at with 98% bit accuracy, while the recent ConvNeXt-v2/UNet VideoSeal hybrid of Fernandez et al. (2024) extends the idea to video, with the most recent version able to hide up to 256 bits. None of those models consider security issues. Different from bit-stream embedding models, Hide-R (Evennou et al., 2024) is a vector-based variant of HiDDeN that embeds continuous representations for the purpose of transmitting messages. In the same spirit, we propose a vector variant of the more robust model VideoSeal.

**Autoencoders** Beyond decoder-only models, modern text autoencoders are typically framed as denoising objectives, where the input is intentionally corrupted, and the model is trained to reconstruct it. Approaches like MASS (Song et al., 2019) and BART (Lewis et al., 2019) have demonstrated that masking or permuting text spans yields latent representations that are beneficial for downstream generation tasks. T5 (Raffel et al., 2020) further unifies this concept into a text-to-text framework. Earlier variational approaches, such as the Neural VAE proposed by Bowman et al. (2016) and its refinement by Shen et al. (2020), explicitly aim to encourage smooth semantic latent spaces. However, these methods often struggle with posterior collapse. Autoencoder approaches never add gaussian noise to the latent space during training, as they optimize for reconstruction accuracy and not for robustness.

**Text Compression** Optimal sentence lossless encoding is achieved through LLMZip (Valmeekam et al., 2023) which leverages an LLM to feed symbol probabilities to an arithmetic encoder in order to craft a losslessly compressed variable-length bitstream.

## 3 METHOD

LatentSeal combines a watermarking model and a robust text autoencoder specifically designed for watermarking purpose (Fig. 1). The watermark model relies on VideoSeal (Fernandez et al., 2024) that reaches a capacity of 256 bits with SOTA performance. We modify the model to embed a 256-D unit-norm real-valued vector. The vector dimension represents a trade-off among capacity, robustness, memory requirements, and computation time; larger vectors typically demand more resources while decreasing robustness. The text autoencoder must satisfy two key constraints: (i) its latent space should be 256-dimension, which is small size for text embedding (ii) it must be robust to the noise introduced by the watermarking model and the transmission. In this context, having an efficient encoder is crucial, and we choose ModernBert (Warner et al., 2024) finetuned using LoRA (Hu et al., 2021) as encoder. We use the [CLS] token as a representation of the input sentence and project it into a 256-D unit-norm space, that constitutes our latent space. For the

decoder, we designed a lightweight transformer architecture which allows fast decoding. Robustness is achieved by introducing gaussian noise into the latent space during training to mimic the impact of the watermark extraction. Despite the challenging constraint of a small latent space, we achieved nearly perfect reconstruction. **The overall architecture necessitates two independent training phases**: one for the watermark model and another for the text autoencoder.

Finally, a key-conditioned reversible rotation is introduced at inference to ensure security in the watermarking process.

### 3.1 WATERMARK EMBEDDER AND EXTRACTOR

As VideoSeal (Fernandez et al., 2024), the embedder is based on a UNet architecture, and ConvNeXt-v2 is used as as watermark extractor. Our version is trained to embed $y \in \mathbb{S}^{255} = \{y \in \mathbb{R}^{256} : \|\{y\}\| = 1\}$ within a cover image $I_c$ of dimension $3 \times 256 \times 256$. The vector embedding occurs in the bottleneck of the embedder (see App. B.1 for the detailed architecture). The watermark model should maximize the similarity between the watermark embedded in the cover $y$ and the one extracted $\hat{y}$, given by the cosine similarity.

During training, we sample the vectors $y^{(n)}$ uniformly from the surface of the unit hypersphere:

$$y'^{(n)} \sim \mathcal{N}(0, I_{256}), \qquad y^{(n)} = \frac{y'^{(n)}}{\|y'^{(n)}\|_2}, \tag{1}$$

Instead of a binary cross-entropy used in Fernandez et al. (2024), our version is trained to minimize:

$$\mathcal{L}_{\text{cosine}}(\hat{y}, y) = 1 - \frac{\hat{y} \cdot y}{\|\hat{y}\|_2 \|y\|_2} \tag{2}$$

Ensuring robustness demands high cosine similarity even under classic image transforms (e.g. colorimetric transforms, crop, JPEG compression). This is achieved by augmentations during training. Imperceptibility requires invisibility from the human eye and is traditionally measured with the PSNR. Alike Fernandez et al. (2024), we use a MSE term in the YUV domain between the input image and the watermarked one. Besides, we use a scheduled factor before the watermarking signal to reduce the perceptibility during the training, making it increasingly harder to keep a high cosine similarity. Hence the model has to learn to deal with a lower intensity signal. More information about training is given in App. B.2.

### 3.2 TEXT AUTOENCODER

The proposed autoencoder learns to map input sequences of 30 tokens onto the surface of the unit hypersphere $\mathbb{S}^{255}$. We find 30 tokens to be aligned with the highest capacity of existing multibit watermarker, as detailed in App. D.3. We use the ModernBert-base (Warner et al., 2024) tokenizer which has a vocabulary size of 50,368. Note that without compression, 30-token sequences correspond to 468 bits of information, given that $log_2(50,368) \approx 15.6$ bits per token. While Evennou et al. (2024) transforms text into vectors in two distinct steps, first compressing with LLMZip and then modulating via Truncated Cyclic Code Shift Keying, our autoencoder streamlines these two steps into a single process, all while significantly increasing the overall capacity.

**Architecture** The encoder $E$ is a pretrained encoder-only model, ModernBERT-base (Warner et al., 2024), on top of which a projection layer maps the [CLS] token to a 256-D unit-norm space. The design of our decoder $D$ is however central to our method. A standard causal decoder is unsuitable because it lacks a mechanism to incorporate the latent vector at each decoding step. We address this by leveraging the memory mechanism of torch.nn.TransformerDecoderLayer, as depicted in Figure 2. Full implementation details are given in App A.1.

In a conventional encoder-decoder transformer architecture (Vaswani et al., 2017), the cross-attention module uses the encoder's output as its key and value tensors. We deviate from this standard approach by not feeding the outputs directly from ModernBERT. Instead, we project the [CLS] token to $\mathbb{S}^{255}$ and, crucially, add noise during the training phase to enhance robustness. This latent vector then serves as the key and value for the decoder's cross-attention module at every decoding step. This critical modification allows the decoder to condition its output on the meaningful

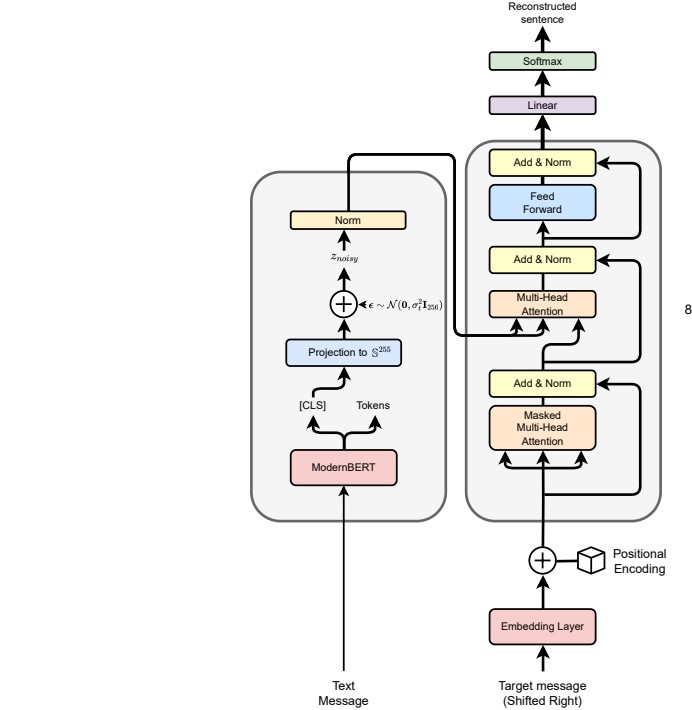

Figure 2: Text autoencoder architecture. Encoder (left) outputs are fed as memory (key,value) to the MHA module of the decoder layers (right).

watermark. The introduction of noise during training simulates perturbations that can occur from image processing or watermark extraction, thereby training the decoder to effectively denoise its input. Our complete autoencoder has 197M parameters, with 155M in the encoder and only 42M in the decoder. The lightweight design of the decoder enables fast decoding, which is beneficial for real-world applications.

**Training** The autoencoder $D \circ E$ is optimized with the standard token-level cross-entropy:

$$\mathcal{L}_{\text{rec}} = -\frac{1}{N_{\text{tok}}} \sum_{t=1}^{N_{\text{tok}}} \sum_{i=1}^{|\mathcal{V}|} \mathbf{q}_i^{(t)} \log\big(\mathbf{p}_i^{t)}\big), \tag{3}$$

where $N_{\text{tok}}$ is the number of non-padded tokens of a sequence, $\mathcal{V}$ denotes the vocabulary and $\mathbf{q}, \mathbf{p} \in \mathbb{R}^{|\mathcal{V}|}$ are respectively the ground-truth and predicted probability vectors. Probabilities are obtained from the decoder logits $\mathbf{z}^{(t)} \in \mathbb{R}^{|\mathcal{V}|}$ with a softmax layer.

### 3.3 TRUSTWORTHINESS AND SECURITY

Leveraging a unit-norm vector allows for the implementation of mechanisms that ensure the watermark's reliability and security during inference.

**Confidence via spherical p-values** We provide a two-step confidence mechanism in decoding at inference. Let $\hat{m} = D(\hat{y})$ be the text decoded from the extracted vector $\hat{y}$. We first apply an idempotence test: the decoded text is passed once more through the text autoencoder and accepted only if $D \circ E(\hat{m}) = \hat{m}$. A different message indicates a deviation from the autoencoder's distribution, suggesting a too strong distortion of the vector $\hat{y}$. In the other case, we compute the cosine sim-

ilarity $c = \cos(\hat{y}, E(\hat{m}))$. A high value indicates the latent $\hat{y}$ remained in the same latent vector neighborhood as $\hat{m}$ and the message is likely correct, whereas a low value flags potential corruption.

Under the null hypothesis that $y$ is a random unit vector sampled in $\mathbb{S}^{255}$, the one-sided $p$-value $\rho$ is given by:

$$\rho = \Pr(\cos(\hat{y}, E(\hat{m})) \geq c) = I_{1-c^2}\left(\frac{255}{2}, \frac{1}{2}\right), \quad (4)$$

where $I_x(a, b)$ denotes the regularised incomplete beta, and $c > 0$. This yields a per-sample surprise score $\rho \in [0, 1]$: small $\rho$ indicates substantial information retention, whereas $\rho \approx 1$ implies independency between $\hat{y}$ and $E(\hat{m})$. In appendix B.3, we provide a detailed theoretical analysis of our system when used for watermark detection.

**Security with secret key-based rotation**  The security of our system is implemented at inference via an inversible rotation system conditioned by a secret key. This is a solution by design, as it takes into accounts both the latent dimension and unit-norm property. This allows the mechanism to be seamlessly integrated post-training without significant computational overhead (see App. C). The secret rotation layer is introduced to provide message confidentiality via a key, in line with standard cryptographic practice (Cox et al., 2008) where security relies on a secret key rather than obscurity of the algorithm. Concretely, a content creator Alice encodes a provenance message $m$ into a latent $y = E(m)$ and then applies a key-conditioned rotation $\mathbf{O} \in \mathrm{SO}(256)$ to obtain $y_r = \mathbf{O}y$, which is embedded into the image. An unauthorized party (Eve) with access to the watermarked image and the public models can only extract a rotated vector $\hat{y}_r$; without the key $k$ she cannot invert $\mathbf{O}$, so $D(\hat{y}_r)$ yields an unintelligible message. By contrast, an authorized verifier (Bob) who knows $k$ reconstructs $\mathbf{O}$, computes $\hat{y} = \mathbf{O}^\top \hat{y}_r$, and recovers $m$ via $D(\hat{y})$. This mechanism is applied post-training, incurs negligible computational overhead, supports arbitrarily many keys, and targets confidentiality rather than robustness, which is handled by the watermarking architecture itself.

## 4 EXPERIMENTS AND RESULTS

For our experiments, we use 2,000 images from the COCO-2017 validation set as cover images, each watermarked with various text samples from one of the datasets described below. Qualitative examples can be found in Fig. 10 of App. F.1.

**Datasets**  We benchmark on three corpora of ascending difficulty. COCO-2017 (Lin et al., 2015) supplies 566 k short captions (11M tokens) and serves as a baseline for simple language. PixMo-Cap (Deitke et al., 2024) provides 700 k image captions (21M tokens); these longer, vocabulary-rich descriptions approximate real image-captioning use. WikiText-103-v1 (Merity et al., 2016) offers 1.8M training samples (48M tokens) plus 4k-document dev/test splits, sourced from Wikipedia. While the two first datasets are related to the image tampering detection application mentioned above, the last one is used for evaluation purposes: its wide thematic diversity and density of named entities stress our system's ability to convey detailed information. Together, the three datasets test robustness from concise, repetitive captions to high-entropy articles.

**Baseline**  We compare LatentSeal against VideoSeal which is the current highest-capacity watermarking model with a 256-bits capacity. Other existing models exhibit significantly lower capacities (see Table 16 in App. B.5), with the best alternative to VideoSeal conveying at most 100 bits (Bui et al., 2023). VideoSeal is fed with text using LLMZip to get a bitstream that serves as watermark. For a fair comparison, both our text autoencoder and VideoSeal use an input length of 30 tokens, which corresponds as closely as possible to a message length of 256 bits obtained by LLMZip (see App. D.3 for this estimation). The resulting bitstream may be truncated if longer.

**Training**  The watermark model and the text autoencoder that make up LatentSeal are trained independently. We deliberately avoid end-to-end training of the watermarker and the text autoencoder. The watermarker is trained to embed and extract random unit vectors on the hypersphere $\mathbb{S}^{255}$, generated by a key-conditioned pseudo-random process, so its behaviour is intentionally independent of any specific upstream encoder and already achieves high fidelity. Jointly training both components would couple the watermarker to the latent distribution of a single encoder, reducing its usefulness

Table 1: Hyperparameter search results for the Text Autoencoder at 30 tokens. Models are sorted by best validation BLEU score within each section. The best performance for each metric across all runs is highlighted in bold. Time is reported in hours.

| Text AE Decoder | | | Training Config | | | Results | | |
|---|---|---|---|---|---|---|---|---|
| Hidden (d) | Layers (N) | Latent (z) | LR | Batch | LoRA r | BLEU4 ↑ | Loss ↓ | Time (h) |
| Convergence Runs (Training Time ≥ 2 hours) | | | | | | | | |
| 512 | 10 | 256 | $2.5 \times 10^{-4}$ | 128 | 32 | **98.72** | **0.0343** | 10.33 |
| 512 | 10 | 256 | $5.0 \times 10^{-4}$ | 256 | 32 | 98.59 | 0.0360 | 6.97 |
| 768 | 8 | 256 | $3.0 \times 10^{-4}$ | 256 | 32 | 97.56 | 0.0556 | 6.72 |
| 512 | 8 | 256 | $3.0 \times 10^{-4}$ | 256 | 32 | 95.44 | 0.0504 | 6.67 |
| 512 | 10 | 256 | $1.5 \times 10^{-4}$ | 128 | 32 | 95.27 | 0.0451 | 10.80 |
| 768 | 12 | 256 | $3.0 \times 10^{-4}$ | 256 | 32 | 94.55 | 0.0436 | 7.34 |
| 768 | 8 | 256 | $1.5 \times 10^{-4}$ | 128 | 32 | 94.75 | 0.0457 | 10.20 |
| 512 | 8 | 256 | $1.5 \times 10^{-4}$ | 128 | 32 | 93.71 | 0.0625 | 10.03 |

as a generic continuous channel for other modalities or encoders, while substantially increasing optimization complexity and the size of the hyperparameter space. In practice, the modular design offers better generality, stability, and reusability, with no clear evidence that end-to-end training would provide benefits. For the watermarking model, we finetune the Videoseal model in order to adapt it to our substantial modifications of input and output spaces. For the text autoencoder, the encoder part comprises ModernBert-Base that we finetune using LoRA at rank 32 on q,k,v layers with $\alpha$ at 64. An hyperparameter search was performed for text autoencoder decoder part (Table 1). Regarding the baseline VideoSeal+LLMZip, the LLMZip used relies on OPT-125M that we finetune using LoRA (Hu et al., 2021).

**Computation** We used up to 4 H100 GPU to train the autoencoder and test diverse architectures. For most evaluation scripts, we use one A40 GPU. Both watermarking and text autoencoder models are rather frugal and do not need a lot of computing power at inference time. The whole pipeline needs less than 3 GB of VRAM with a batch size of 1 to run.

**Security** Following (Mezzadri, 2007), we draw a rotation matrix uniformly at random with a key $k$. Both sampling and applying the rotation are cheap, about 1ms, ensuring security at a low cost. Details on implementation and computational cost are given in App. C.

**Metrics** We assess reconstruction quality with the BLEU-4 metric (Papineni et al., 2002), which accounts for n-gram precision between a candidate and a reference sentence. Moreover, we use the exact match rate (EM) to measure the rate of messages perfectly reconstructed. We report speed as the number of tokens processed per second.

### 4.1 WATERMARKING MODEL ROBUSTNESS AND PERCEPTIBILITY

Table 2 shows the impact of valuemetric and geometric transformations on the watermark. We obtain significant improvements in terms of BLEU4 metric over a wide range of transformations. We unsurprisingly conserve high cosine similarity as most of the transformations were included while training the watermarking model. Watermark perturbation is measured by PSNR between original and watermarked images, we aim at 42dB in our experiments as we require high-standard imperceptiblity.

To target a specific PSNR at inference, we modulate the watermarking perturbation $w$ with $\alpha$ :

$$w = \alpha |I_w - I_c|, \tag{5}$$

where $I_c$ is the cover image, $I_w$ results from a forward pass of the watermark embedding model and $\alpha$ being the scaling factor to control the watermark strength. To study the impact of $\alpha$, we perform a sweep in Table 3 across 2,000 images from the COCO-2017 validation set.

This table shows that $\alpha = 0.2$ achieves a favorable trade-off: PSNR $\approx$ 42 dB with SSIM $\geq 0.989$ and low LPIPS, while preserving high cosine similarity of the recovered message even under common distortions. For our experiments, we use this parameter at $\alpha$=0.2 to target 42 dB of PSNR. We provide an additional robustness study at different PSNR targets in Table 15, App. B.4.

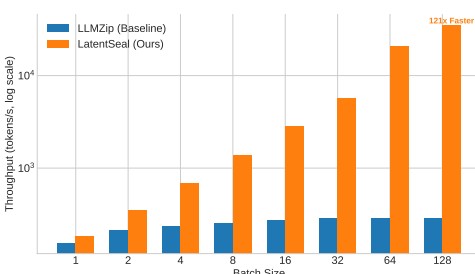
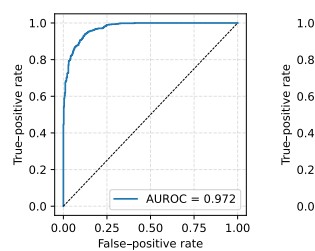
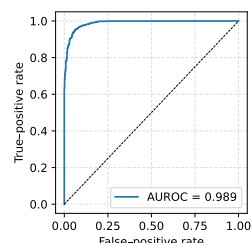

Figure 3: AE decoding speed vs LLMZip ; we report the number of tokens per second.

Figure 4: ROC of $\ell = -\log_{10}\rho$ on non-catastrophic transforms on Wikitext/PixMo-Cap.

## 4.2 TEXT AUTOENCODER

**Speed** Our model achieves significantly higher decoding throughput than OPT-125M (e.g., $121\times$ faster at batch size 128), as shown in Fig. 3. This is due to three key factors: it attends to a fixed-length latent vector $y$ instead of a growing prefix, avoiding quadratic attention costs; it does not require key/value caching during generation, reducing memory usage and access overhead; and its decoder operates with smaller hidden dimensions (512 against 768) and fewer parameters per layer, making each step cheaper.

Table 2: Robustness comparison on PixMo-Cap. Best results in **bold**. All differences are significant at $p < 0.05$ (Wilcoxon).

| Transform | LatentSeal | | VideoSeal+LLMZip | |
|---|---|---|---|---|
| | BLEU-4 | EM(%) | BLEU-4 | EM(%) |
| identity | **97.34** | **92.84** | 92.97 | 89.20 |
| *Geometric* | | | | |
| HFlip | **97.15** | **92.23** | 93.52 | 89.40 |
| Rot 5 | **95.08** | **87.89** | 89.67 | 84.36 |
| Rot 10 | **88.26** | **74.17** | 79.82 | 71.14 |
| Rot 90 | **94.54** | **87.39** | 90.66 | 85.67 |
| Crop 90 | **93.27** | **83.50** | 80.14 | 71.75 |
| Persp 0.3 | **85.01** | **69.88** | 77.26 | 67.66 |
| Persp 0.5 | **36.99** | **10.99** | 9.69 | 0.96 |
| *Noise/Blur* | | | | |
| GNoise 10 | **13.45** | 0.00 | 1.83 | 0.00 |
| GBlur 5 | **8.75** | **0.55** | 1.17 | 0.00 |

| Transform | LatentSeal | | VideoSeal+LLMZip | |
|---|---|---|---|---|
| | BLEU-4 | EM(%) | BLEU-4 | EM(%) |
| *Color/Contrast* | | | | |
| Sat 0.5 | **96.74** | **92.18** | 93.14 | 89.10 |
| Sat 1.5 | **96.86** | **92.13** | 92.59 | 88.14 |
| Cont 0.5 | **95.84** | **90.41** | 92.69 | 88.60 |
| Cont 1.5 | **84.95** | **70.23** | 72.96 | 64.18 |
| Bright 0.5 | **95.95** | **90.36** | 91.75 | 87.18 |
| Bright 1.5 | **83.48** | **69.93** | 72.00 | 63.72 |
| Hue 0.1 | **92.72** | **82.34** | 79.53 | 71.39 |
| Hue -0.1 | **92.32** | **81.58** | 79.88 | 71.64 |
| *JPEG Compression* | | | | |
| Jpeg 80 | **89.39** | 74.27 | 86.08 | **78.71** |
| Jpeg 75 | **85.60** | 68.06 | 81.29 | **71.49** |
| Jpeg 70 | **81.21** | 58.83 | 75.68 | **63.87** |
| Jpeg 60 | **66.50** | 31.48 | 51.99 | **33.75** |
| Jpeg 50 | **47.56** | 10.95 | 31.61 | **13.82** |
| Jpeg 40 | **24.32** | 0.45 | 11.12 | **1.56** |

**Reconstruction performance** In Table 4, we exhibit state-of-the-art performances across both COCO-2017 and PixMo-Cap. We highlight a better BLEU4 under all transformations, suggesting better global reconstruction. The Exact Match (EM) rate, a strict binary metric, shows similar results except under JPEG compression at 75% quality. However, the significantly higher BLEU4 score indicates that while our method may miss a few tokens for perfect reconstruction, it is globally able to reconstruct the message more effectively with graceful degradation, see 14. We offer additional results in the $> 256$ bits regime in D.1 and evoke possible test set leakage in LLMZip.

**Confidence score** The confidence score $\rho$ measures how self-consistent a latent is after a decode-encode round-trip. For convenience, we consider $\ell = -\log_{10}\rho$. Larger $\ell$ means *higher* confidence, and we can set a threshold $\ell_{th}$ such that only messages with $\ell > \ell_{th}$ should be trusted. A threshold $\ell_{th} = 109$ therefore rejects captions whose p-value exceeds $10^{-109}$. We provide ROC curves in Fig. 4 outlining the ability of this confidence metric to filter out damaged watermarks. As Table 5 shows, setting $\ell_{th} = 154.94$ keeps 77.9 % of perfect reconstructions while discarding 99 % of

Table 3: Scaling factor $\alpha$ sweep and corresponding perceptibility metrics with cosine similarity between embedded watermarks and extracted ones. We keep $\alpha$=0.2 for further experiments.

| | Perceptibility metrics | | | Cosine similarity | | | | |
|---|---|---|---|---|---|---|---|---|
| $\alpha$ | PSNR↑ | SSIM↑ | LPIPS↓ | Clean | JPEG-50 | JPEG-30 | Noise | Crop |
| 0.05 | 49.7 | 0.9982 | 0.0008 | 0.7581 | 0.5537 | 0.4068 | 0.6152 | 0.6132 |
| 0.10 | 47.14 | 0.9967 | 0.0017 | 0.9250 | 0.8193 | 0.6910 | 0.8567 | 0.8658 |
| 0.15 | 44.43 | 0.9940 | 0.0031 | 0.9697 | 0.9235 | 0.8507 | 0.9394 | 0.9452 |
| **0.20** | **42.20** | **0.9897** | **0.0055** | **0.9837** | **0.9601** | **0.9200** | **0.9677** | **0.9693** |
| 0.25 | 40.41 | 0.9844 | 0.0088 | 0.9877 | 0.9726 | 0.9465 | 0.9774 | 0.9774 |
| 0.30 | 38.92 | 0.9780 | 0.0134 | 0.9886 | 0.9778 | 0.9591 | 0.9815 | 0.9798 |
| 0.35 | 37.66 | 0.9713 | 0.0186 | 0.9881 | 0.9790 | 0.9635 | 0.9819 | 0.9794 |
| 0.40 | 36.55 | 0.9624 | 0.0263 | 0.9868 | 0.9800 | 0.9675 | 0.9817 | 0.9778 |

Table 4: VideoSeal vs. LatentSeal (Ours) comparison. PixMo-Cap/COCO-2017 results are at 42dB with 30 tokens. Wikitext results are for messages $> 256$ bits. Best result for each metric is in **bold**. We exhibit state-of-the-art performances on all settings. BLEU4 results for VideoSeal on WikiText are biased, as the method truncates above 256 bits sentences.

| | | PixMo-Cap | | COCO-2017 | | Wikitext ($> 256$ bits) | |
|---|---|---|---|---|---|---|---|
| Transf. | Method | EM (%) | B4 | EM (%) | B4 | EM (%) | B4 |
| Identity | VideoSeal+LLMZip | 89.2 | 93.0 | 96.9 | 97.4 | 0 | **79.9** |
| | LatentSeal (Ours) | **92.8** | **97.3** | **97.7** | **99.0** | **51.6** | 76.3 |
| JPEG-75 | VideoSeal+LLMZip | **71.5** | 81.3 | 85.1 | 87.4 | 0 | **64.4** |
| | LatentSeal (Ours) | 68.1 | **85.6** | **85.2** | **93.9** | **7.9** | 39.3 |
| Rot (5°) | VideoSeal+LLMZip | 71.1 | 79.8 | 93.9 | 94.7 | 0 | **77.3** |
| | LatentSeal (Ours) | **87.9** | **95.1** | **94.6** | **97.6** | **40.2** | 65.7 |
| Crop (90%) | VideoSeal+LLMZip | 71.8 | 80.1 | 87.9 | 89.9 | 0 | **69.5** |
| | LatentSeal (Ours) | **83.5** | **93.27** | **90.7** | **95.2** | **30.5** | 59.1 |
| Noise | VideoSeal+LLMZip | 0.0 | 1.8 | 0.3 | 0.0 | 0.0 | 0.7 |
| | LatentSeal (Ours) | 0.0 | **13.5** | **0.8** | **17.3** | 0.0 | **4.5** |

erroneous ones for PixMo-Cap. For safety-critical deployment a stricter $\ell_{min} = 170.16$ rejects every untrustworthy watermarks yet retains 63.4% of reliable messages.

## 4.3 ROBUSTNESS TO IMAGE EDITING MODELS

To evaluate if Videoseal and LatentSeal are robust to attack from image editing models, we modify images with HiDream (Cai et al., 2025) and IP2P (Zhang et al., 2023) and report latent vector similarity between original and decoded messages using SBERT and BLEU4 in Fig. 8 of App. E. We found dramatic degradation for both methods, which is not surprising given recent work on Image Regeneration (Liu et al., 2025) that uncovered this vulnerability. Nevertheless, we compare favorably with Videoseal to that regard and, thanks to our confidence metric, are able to flag at inference time if the message is likely tampered with or not. Also, PSNR between original and edited images have a median of 14.9 for HiDream and 19.7 for IP2P. These perturbations thus dramatically disrupt image quality, demonstrating that any attempt to remove the watermark comes at the cost of degrading the image's quality. More details are provided in App. E.

## 4.4 ABLATION STUDY

To probe the robustness and limitations of our design, we conduct targeted ablations on three key components of the system: encoder capacity, bottleneck size, and distribution shift.

Table 5: Operating points for Fig. 4. A received message is *untrustworthy* (negative) if $\ell < \ell_{th}$.

| Dataset | Target FPR | Threshold $\ell_{th}$ | TPR |
|---|---|---|---|
| PixMo-Cap | $10^{-4}$ | 170.16 | 63.4 % |
| | $10^{-2}$ | 154.94 | 77.9 % |
| | $10^{-1}$ | 120.66 | 97.2 % |
| Wikitext | $10^{-4}$ | 192.69 | 44.8 % |
| | $10^{-2}$ | 176.84 | 61.9 % |
| | $10^{-1}$ | 139.75 | 90.7 % |

**Encoder capacity**    On Wikitext, we swap the ModernBERT-base encoder for a larger ModernBERT-large (same hyper-parameters). BLEU4 plateaus from 98.0 to 97.5, suggesting that the encoder does not impact the reconstruction accuracy, likely due to the bottleneck dimension.

**Latent dimensionnality**    We study the effect of the autoencoder latent dimensionality $z$ in a dedicated sweep over $\{128, 256, 384\}$ in Table 8. 256 dimensions provide the best trade-off between reconstruction quality and robustness to the noise injected in the latent space during training. In contrast, models with 128 and 384 dimensions stagnate earlier and consistently underperform the 256-dimensional model, despite exploring multiple architectures and hyperparameters. Since the text autoencoder is the bottleneck of the overall system, we therefore adopt $z = 256$ in all main experiments. The watermarking component itself can in principle operate with other latent sizes.

**Bottleneck size**    Removing the 256-D bottleneck and keeping the 768-D of [CLS] prevents convergence; the model does not converge and struggles at 0.3 BLEU4. Hence a compact latent space is *necessary* for stable training.

**Data Novelty**    On PixMo-Cap, we gather a challenging test set, PixMo-Cap-C, that comprises the top 5 % hardest samples, based on the *4-gram Novelty Score* for a sentence $S$ defined as:

$$\text{Novelty}(S) = \frac{|\{g \in G_4(S) \mid g \notin \mathcal{G}_{\text{train}}\}|}{|G_4(S)|},$$

where $G_4(S)$ is the multiset of 4-grams in $S$ and $\mathcal{G}_{\text{train}}$ is the set of 4-grams in the training corpus (sentences too short to contain a 4-gram are excluded). When evaluating on PixMo-Cap-C, BLEU4 drops modestly ($99.1 \rightarrow 94.2$), indicating generalization remains strong.

## 5    CONCLUSION, LIMITATIONS, AND FUTURE WORK

LatentSeal demonstrates that watermarking need not be constrained to bit payloads. By embedding autoencoded text vectors, we move from raw capacity numbers to a semantic channel that transmits meaningful, verifiable, and secure information. While we focus on English text, making our models not suited for other languages, experiments on COCO, PixMo-Cap, and WikiText confirm that this semantic channel achieves higher fidelity and robustness than bit-based baselines, while decoding 120× faster than existing pipelines.

Equally important, our keyed latent rotations provide a lightweight form of cryptographic security, and our confidence metric offers statistical guarantees of reliability. Together, these features make watermarking not just robust, but trustworthy at deployment scale.

Even if image editing models can strip our watermark (not addressed here), we use it as a trustworthiness flag. Its absence prompts additional forensic scrutiny.

We did not study the scale of the decoder due to compute limitation and the bottleneck of the latent dimension. This is a design choice for real-world use, but, as it is often the case in deep learning, better reconstruction might be achieved using deeper models, at the price of inference latency.

We see this as the start of a broader research direction: watermarking as semantic communication. Extending latent space watermarking to video, audio, or cross-modal provenance could enable practical authentication pipelines for the AI era, where watermarks no longer just exist, but *explain*.

## REPRODUCIBILITY STATEMENT

Section 3 provides the specifics of our method and training. Implementation is fully detailed in App A and App B, including model architectures, hyperparameters, and dataset splits. PyTorch implementations are given as supplementary material, together with training and inference code. They will be released publicly upon publication. All experiments are run with fixed seeds to ensure determinism. We rely exclusively on publicly available datasets, and we will release the exact data splits used in this paper. A complete list of software dependencies and environment specifications will also be provided to enable faithful replication of our results.

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

# A AUTOENCODER DETAILS

All code is available for review in the zip supplementary. Checkpoints will be released upon publication.

## A.1 ARCHITECTURE

Given a tokenized input $x = (x_1, \ldots, x_T)$, a ModernBERT encoder $f_\theta$ produces $H \in \mathbb{R}^{T \times d_e}$. We pool with the [CLS] vector to obtain $h \in \mathbb{R}^{d_e}$, then project with $W \in \mathbb{R}^{d_e \times d}$ to $z' = Wh \in \mathbb{R}^d$, followed by $\ell_2$-normalization $z = z'/\|z'\|_2 \in \mathbb{S}^{d-1}$ (here $d = 256$). Optional isotropic noise $\varepsilon \sim \mathcal{N}(0, \sigma^2 I)$ of variance $\sigma^2$ is added before re-normalization.

Decoding uses a Transformer decoder with $L = 10$ layers, $H = 8$ heads, model width $d = 256$, FFN width $4 \cdot 512 = 2048$, dropout $0.01$, and sinusoidal positional embeddings (position 0 zeroed); target embeddings are scaled by $\sqrt{d}$ and weights are tied ($W_{\text{out}} = E^\top$). The latent $z$ is broadcast as time-invariant memory, and next-token cross-entropy under teacher forcing (pad ignored) is optimized with Adam and a cosine scheduler. Generation defaults to greedy decoding up to $T$ tokens.

ModernBERT is adapted with LoRA rank=32 and all non-LoRA backbone weights are frozen; the projection, decoder stack, embeddings, and LoRA adapters are trainable. Evaluation reports BLEU4.

Table 6: Autoencoder architecture with implementation details.

| Component | Symbol / Shape | Implementation details |
|---|---|---|
| Vocabulary | $V$ | 50,368 (From ModernBert tokenizer) |
| Encoder | $H \in \mathbb{R}^{T \times d_e}$ | ModernBERT-base; pooling by picking [CLS]. |
| Latent projection | $W \in \mathbb{R}^{d_e \times d}$ | Linear to $d = 256$, then $\ell_2$-normalize to $\mathbb{S}^{255}$. Optional noise $\sigma$ for robustness. |
| Target embedding | $E \in \mathbb{R}^{V \times d}$ | Learned; output layer tied: $W_{\text{out}} = E^\top$. |
| Positional encoding | $P \in \mathbb{R}^{T \times d}$ | Sinusoidal, length $30 + 64$; $P_{0,:} = 0$; dropout $0.01$; token scale $\sqrt{d}$. |
| Decoder | TransformerDecoder | $L = 10$ layers, $H = 8$ heads, $d = 256$, FFN $= 2048$, dropout $0.01$; causal mask; memory is $z$ broadcast to length $T$. |
| Special tokens | BOS/EOS/PAD | BOS from [CLS] (fallback 0); EOS from [EOS] or [SEP]; PAD from tokenizer (fallback 0). |
| Objective | $\mathcal{L}_{\text{NLL}}$ | Next-token cross-entropy with teacher forcing; padding ignored. |
| Optimization | Adam + cosine | $\texttt{lr} = 5 \cdot 10^{-4}$; CosineAnnealingLR with $T_{\max} = 100$, $\eta_{\min} = 10^{-6}$. |
| Decoding | greedy | Greedy by default up to $T$ |
| Evaluation | BLEU4 | Reported at validation/test epoch end. |

## A.2 TRAINING

**Dataset splits** Details on the sets used for the autoencoder training and LatentSeal evaluation (Test sets) are given in Table 7 for each dataset. The train/val/test set splits are the original ones given for WikiText. For COCO-2017, our validation and test set result from splitting the original validation set. For PixMo-Cap, we randomly split the given dataset. Our custom splits will be released with the code.

**Text samples** We construct the 30 tokens text samples by truncation of dataset samples. This may introduce a bias to toward better representing the start of sentences but as it is aligned with our goal, we do not find it hampers our method. Note that for comparison purpose, we opt for a less wasteful method with the 700k training samples of PixMo-Cap. As samples maybe composed of several sentences, we tokenize each sample with ModernBert's tokenizer and greedily pack full

Table 7: Dataset statistics. We use custom splits for **PixMo-Cap** and **COCO-2017** validation/test.

| Dataset | Train | Validation | Test |
|---------|-------|------------|------|
| COCO2017 | 300k | 10k | 15k |
| PixMo-Cap | 3,971,255 | 44,125 | 397,126 |
| WikiText | 1.8M | 3,760 | 4,760 |

sentences until adding the next sentence would exceed $N$ tokens. Whenever adding the next piece would exceed $N$, we start a new sample. If a single sentence is longer than $N$, we split that sentence into clauses at commas, semicolons, or colons, and pack those the same way, and pack clauses the same way. The result is linguistically coherent training samples that are each $\leq N$ tokens, favoring intact sentences over exact-$N$ lengths. This process also enables to increase the number of samples, going from the original 700k training samples to almost 4M of 30-tokens samples. Note that we do not observe performance gap between this method and a more naive hard truncation at $N$ tokens.

**Latent Noising**   We find that the maximum noise at which the AE still converges to high level of BLEU4 is obtained with a $\sigma = 0.01$ to sample Gaussian Noise.

**Hyperparameter optimization**   Table 8 shows the complete hyperparameter search used in our experiments to train the autoencoder.

### A.3   ADDITIONAL EXPERIMENTS

**Encoder outputs**   Two main possibilities can be used about the output of the encoder, either pooling or just selecting the [CLS] token. In our experiments, we find subpar performances when pooling, either averaged or learned ones.

**Impact of the number of tokens**   To showcase the impact of the number of tokens, we evaluate the viability of our system to encode/decode up to 40 tokens, using the same robustness to a noise of $\sigma = 0.01$. Results are given in Table 9, and indicate that performance gracefully degrades as sentence size increase.

**Random token ids**   We emphasize that our text autoencoder is trained on natural-language sentences and is therefore designed to represent *semantic* text, not arbitrary identifiers. To probe its behaviour on out-of-distribution inputs, we conduct a simple experiment where we sample 1000 random "sentences" by drawing 30 token IDs uniformly from the ModernBERT vocabulary, encode them into the 256-dimensional latent space, and decode them back to text. We then compute exact string reconstruction and BLEU-4 between the input and decoded sequences. Over this random-ID set, the exact-match rate is $0\,\%$ and the average BLEU-4 score is $\approx 0.0029$, indicating that none of the decoded outputs reproduces the input token sequence. Qualitatively, the decoder instead produces linguistically plausible fragments that lie on the natural-language manifold but are unrelated to the original random IDs. This confirms that the autoencoder behaves as a semantic compressor rather than as a mechanism for reliably encoding and recovering arbitrary identifier strings.

**Qualitative example (random IDs through the autoencoder).**

**Input (30-token random IDs)**
```
"auseunciation Campbell properties
talked competency technique
PVdecor leur preferred attachment
boundary protocols 701Lin"
```

**Decoded output**
```
"083ovich064pay lawsuit hypers
phenomenalshots Resolution L
twelve let Tigers minor sadly
fuss strength uphold applications
faster Black Nova"
```

Table 8: Hyperparameter search for the Text Autoencoder at 30 tokens on Wikitext. Models are sorted by Best Validation BLEU4 score within each section. The best performance for each metric across all runs is highlighted in bold. Time is reported in hours. Those are the top results, we did perform as many experiments with 384 than with 128 but they were worse.

| Architecture | | | Training Config | | | Results | | |
|---|---|---|---|---|---|---|---|---|
| Hidden (d) | Layers (N) | Latent (z) | LR | Batch | LoRA r | BLEU ↑ | Loss ↓ | Time (h) |
| Convergence Runs (Training Time ≥ 2 hours) | | | | | | | | |
| 512 | 10 | 256 | 2.5e-4 | 128 | 32 | **98.72** | **0.0343** | 10.33 |
| 512 | 10 | 256 | 5.0e-4 | 256 | 32 | 98.59 | 0.0360 | 6.97 |
| 768 | 8 | 256 | 3.0e-4 | 256 | 32 | 97.56 | 0.0556 | 6.72 |
| 512 | 8 | 256 | 3.0e-4 | 256 | 32 | 95.44 | 0.0504 | 6.67 |
| 512 | 10 | 256 | 1.5e-4 | 128 | 32 | 95.27 | 0.0451 | 10.80 |
| 768 | 12 | 256 | 3.0e-4 | 256 | 32 | 94.55 | 0.0436 | 7.34 |
| 768 | 8 | 256 | 1.5e-4 | 128 | 32 | 94.75 | 0.0457 | 10.20 |
| 512 | 8 | 256 | 1.5e-4 | 128 | 32 | 93.71 | 0.0625 | 10.03 |
| Exploratory Sweep (Training Time ≤ 2 hours) | | | | | | | | |
| 512 | 6 | 384 | 5.0e-4 | 256 | 32 | 82.22 | 0.1936 | 1.74 |
| 1024 | 6 | 256 | 5.0e-4 | 256 | 16 | 80.98 | 0.1571 | 1.71 |
| 512 | 4 | 256 | 5.0e-4 | 256 | 8 | 71.27 | 0.2857 | 1.62 |
| 512 | 4 | 256 | 5.0e-4 | 256 | 32 | 70.84 | 0.2746 | 1.62 |
| 512 | 4 | 256 | 3.0e-4 | 256 | 16 | 67.45 | 0.3617 | 1.69 |
| 512 | 6 | 384 | 2.0e-4 | 512 | 32 | 63.94 | 0.3871 | 1.54 |
| 512 | 4 | 256 | 6.0e-4 | 512 | 8 | 63.76 | 0.3767 | **1.41** |
| 768 | 4 | 256 | 1.0e-3 | 512 | 16 | 61.47 | 0.3987 | 1.42 |
| 1024 | 10 | 256 | 2.0e-4 | 512 | 16 | 53.53 | 0.4758 | 1.62 |
| 512 | 8 | 128 | 3.0e-4 | 256 | 8 | 43.62 | 0.4110 | 1.71 |
| 1024 | 8 | 256 | 1.0e-4 | 256 | 8 | 43.65 | 0.6521 | 1.78 |
| 768 | 4 | 256 | 2.0e-4 | 512 | 16 | 43.65 | 0.6960 | 1.43 |
| 768 | 4 | 128 | 5.0e-4 | 256 | 8 | 40.86 | 0.4831 | 1.60 |
| 768 | 6 | 128 | 6.0e-4 | 512 | 32 | 40.38 | 0.4886 | 1.44 |
| 768 | 4 | 128 | 3.0e-4 | 256 | 16 | 29.02 | 0.9398 | 1.60 |
| 1024 | 4 | 128 | 1.0e-4 | 256 | 8 | 17.88 | 2.0437 | 1.60 |
| 512 | 6 | 256 | 1.0e-3 | 512 | 32 | 3.18 | 3.6489 | 1.46 |
| 1024 | 10 | 128 | 6.0e-4 | 512 | 16 | 0.04 | 4.0135 | 1.52 |
| 1024 | 8 | 256 | 1.0e-3 | 512 | 16 | 0.05 | 3.8204 | 1.53 |
| 512 | 8 | 256 | 1.0e-3 | 512 | 16 | 0.04 | 3.6161 | 1.50 |

Table 9: BLEU4 scores at different sentences size.

| | PixMo-Cap | WikiText |
|---|---|---|
| 30 tokens | 99.1 | 98.7 |
| 40 tokens | 95.0 | 89.6 |

# B  WATERMARK EMBEDDER-EXTRACTOR DETAILS

## B.1  ARCHITECTURE

The embedder is a compact U-Net that operates in YUV space to minimize perceptual drift on luminance. An encoder–decoder with skip connections downsamples frames through three stride-2 stages to a bottleneck where the binary message is fused (via small conv/MLP layers). The decoder upsamples back to full resolution, concatenating encoder features at each level, and produces a residual in YUV that is added to the input. A differentiable quantization step on the YUV residual stabilizes training and robustness under codecs and common distortions, consistent with the repo's joint training recipe that injects transforms between embedder and extractor passes.

Table 10: Training hyperparameters (paths, workers, and nbits omitted). Grouped for readability. We follow the naming convention of VideoSeal for the hyperparameters.

| Category | Hyperparameter | Value |
|---|---|---|
| **Data** | video dataset | none |
| | image dataset | coco |
| **Models** | extractor_model | convnext_tiny |
| | embedder_model | unet_small2_yuv_quant |
| **Capacity** | hidden_size_multiplier | 1 |
| | batch_size | 64 |
| | disc_in_channels | 1 |
| **Schedule** | epochs | 601 |
| | iter_per_epoch | 1000 |
| | scaling_w_schedule | Cosine, scaling_min=0.2, start_epoch=200, epochs=200 |
| | scheduler | CosineLRScheduler, lr_min=1e-6, t_initial=200, warmup_lr_init=1e-8, warmup_t=20 |
| **Optimization** | optimizer | AdamW, lr=5e-4 |
| | seed | 0 |
| **Loss & Weights** | lambda_dec | 1.0 |
| | lambda_d | 0.1 |
| | lambda_i | 0.1 |
| | perceptual_loss | yuv |
| | attenuation | jnd_1_1 |
| **Augmentations & Disc** | num_augs | 2 |
| | augmentation_config | configs/all_augs.yaml |
| | disc_start | 50 |
| | scaling_w / scaling_i | 1.0 / 1.0 |

The extractor uses a ConvNeXtV2-Tiny backbone: a 4× stem conv followed by four stages with depths 3-3-9-3 and channel dims 96/192/384/768. Each stage stacks residual blocks with depthwise 7×7 conv, channels-last LayerNorm, 4C MLP with GELU and Global Response Normalization, and DropPath; spatial downsampling is done by three stride-2 convs between stages.

**Single-frame detection with a latent on $S^{255}$.** Given a unit-norm message $y \in S^{255}$, the embedder injects $y$ at the bottleneck and produces a low-amplitude YUV residual that is added to the input frame to obtain the watermarked image. At test time, the extractor uses a ConvNeXtV2–Tiny backbone (4× stem followed by stages with depths 3–3–9–3 and channel widths 96/192/384/768) and, under the gaussian setting, regresses a continuous vector $\hat{y}' \in \mathbb{R}^{256}$ from this single frame rather than per-bit logits. We then project $\hat{y}'$ onto the unit sphere and obtain our extracted vector $\hat{y}$.

## B.2 TRAINING

We finetune VideoSeal using adapted training scripts from Fernandez et al. (2024). We use the available 256-bits checkpoint as a starting point and finetune it with our new objective. Table 10 lists the hyperparameters we used. Furthermore, we use effectively 64,000 images from the train set of COCO-2017 during this finetuning.

For robustness purposes, we use data augmentation at training time. We list them in Figure 11.

Table 11: Data augmentations and their parameter ranges used during training.

| Augmentation | Min | Max |
|---|---|---|
| Resize (scale factor) | 0.7 | 1.5 |
| Crop (relative size) | 0.5 | 1.0 |
| Rotate (degrees) | $-10$ | 10 |
| Perspective (distortion) | 0.1 | 0.5 |
| JPEG quality | 40 | 60 |
| Gaussian blur (kernel size) | 3 | 17 |
| Median filter (kernel size) | 3 | 3 |
| Brightness factor | 0.5 | 2.0 |
| Contrast factor | 0.5 | 2.0 |
| Saturation factor | 0.5 | 2.0 |
| Hue shift | $-0.1$ | 0.1 |

### B.3 THEORETICAL ANALYSIS

As requested, we compare standard methods on the watermark detection task. While they work on different domains and trained for different tasks, we use the $p$-value interface as a common ground for comparison. This provides an outline of each method respective statistical guarantees.

We frame it as a statistical detection test. We consider the null hypothesis $H_0$ that each bit of the output binary message $\hat{m}$ is independent and distributed as a Bernoulli variable with probability of success 0.5, and the alternative hypothesis $H_1$ which is that $\hat{m} = m$.

Given an observed bit accuracy bit accuracy$(m, \hat{m})$, the $p$-value is the probability of observing a bit accuracy at least as extreme as the one obtained under the null hypothesis. It is given by the cumulative distribution function of the binomial distribution:

$$p\text{-value}(m, \hat{m}) = \sum_{k \geq n_{\text{bits}}p} \binom{n_{\text{bits}}}{k}(1/2)^{n_{\text{bits}}} = I_{1/2}(n_{\text{bits}}p, n_{\text{bits}}(1-p) + 1), \quad (6)$$

, where $p$ is the bit accuracy between $m$ and $\hat{m}$ and $I_x(a, b)$ the regularized incomplete beta function.

The above equation, borrowed from Fernandez et al. (2024), enables to compare multibit watermarking methods with different capacity (using the $n_{bits}$ factor).

Note that all those methods are restricted by their operating space. At most, one can hide 256 bits and thus, assuming $bit_{acc}$ at 1.0, reach a $p$-value $= 1e - 77$ derivated from $2^{-256}$.

In order to compare with LatentSeal which operates on $\mathbb{S}^{255}$, we use the following context; under the null hypothesis that $y$ is a random unit vector sampled from $\mathbb{S}^{255}$, the one-sided $p$-value $\rho$ is given by:

$$\rho = \Pr(\cos(y, \hat{y}) \geq c) = I_{1-c^2}\left(\frac{255}{2}, \frac{1}{2}\right), \quad (7)$$

Eq. 7 is quite alike Eq. 4, but the test differs. In Eq. 7 the test is about the probability of $y$ and $\hat{y}$ being this close under the null hypothesis, this does theoretically test the watermark presence. This framing allows us to compute an **unbounded** $p$-value, albeit limited by the numerical precision of the computation as $c \to 1.0$ when $y$ and $\hat{y}$ get closer.

In order to establish empirically a statistical mapping of the watermark landscape, we report in Tab. 12 $p$-values from standard methods and LatentSeal using 1000 images from COCOVal2017 Lin et al. (2015). Moreover, in Tab. 13, we offer a comparative study against VideoSeal at extreme False Positive Rate (below $1e - 50$) and observe better performance. Using vectors from the autoencoder latent space does not degrade performances, this is expected as the core property of lying on the unit hypersphere $\mathbb{S}^{255}$ is respected. This would be different is we had trained in an end-to-end fashion, we would expect better performance for AE latent vectors, at the cost of degrading the "random" setting and performing a time-consuming training.

In summary, LatentSeal not only allows for the transmission of semantic messages but does so while providing stronger statistical guarantees than the best available discrete multibit baselines.

Table 12: Comparison of $p$-values between standard methods in watermaking. All multibits methods are bounded at $p-values = 1e-77$ due to their operating space which allows for at most 256 bits.

| Method | HiDDeN | MBRS | TrustMark | Stable Sig. | VideoSeal | LatentSeal (Ours) |
|---|---|---|---|---|---|---|
| Identity | 14.2 | 70.6 | 29.9 | 13.6 | 76.4 | **191.4** |
| Valuemetric | 10.8 | 59.8 | 27.4 | 12.5 | 73.3 | **161.1** |
| Geometric | 5.5 | 3.3 | 8.5 | 9.8 | 74.8 | **164.9** |
| Compression | 14.2 | 69.9 | 29.7 | 7.5 | 72.5 | **132.2** |
| Combined | 2.6 | 0.4 | 0.8 | 9.3 | 60.9 | **91.0** |

Table 13: TPR at Extreme FPR Thresholds (Averaged). We observe no significative difference from embedding random vectors from $\mathbb{S}^{255}$

| FPR | VideoSeal | LatentSeal (Random) | LatentSeal (Encoded) |
|---|---|---|---|
| $10^{-50}$ | 94.3% | 96.3% | 96.2% |
| $10^{-77}$ | 60.9% | 88.5% | 88.5% |
| $10^{-80}$ | 0% (Bit-depth Limit) | 87.6% | 87.5% |
| $10^{-100}$ | 0% | 79.2% | 79.2% |

### B.4 ROBUSTNESS

We study the robustness of our watermark model through the lens of the cosine similarity. In Table 15, we showcase different type of transforms, encompassing weak to strong ones, and their impact on the similarity of the extracted watermark under different signal amplitudes, ranging from 41db to 43db. We show strong robustness to all of these attacks, which is expected as they are used during training. This is not an issue as these transforms encompass most of those performed on images on the web (which are mostly JPEG or Crop).

**Graceful degradation** is a crucial feature enabled by working in a continuous latent space, as it ensures that noise introduced via a channel does not result in a hard failure of message extraction. Instead of a fixed, arbitrary number of message bits, which would require a perfect, error-free channel for complete reconstruction, the autoencoder learns to map sentences to a continuous 256-D space that is robust to noise. This design means that even if the channel is corrupted, the system can still reconstruct a sentence that is semantically close to the original, provided the input is in-domain. Tab. 14 demonstrates this by showing how the proposed method, LatentSeal, consistently achieves a much higher SBERT score (semantic similarity) compared to the baseline, VideoSeal+LLMZip, resulting in a substantial improvement, especially under heavy attacks like GaussianBlur and Jpeg compression.

### B.5 RELATED ARCHITECTURES

Table 16 introduces existing watermarking methods, their capacity and their perceptibility. We pick VideoSeal as baseline as it is the only model with enough capacity.

## C ROTATION WITH SECRET KEY

Our system's security relies on an inversible rotation system that accounts for both the latent dimension and unit-norm property. By doing so, we can introduce this system after training and without much overhead.

---

[0]StableSignature and RAW primarily report image quality via FID/CLIP scores at matched settings rather than a fixed PSNR value; we therefore do not list a PSNR number here.

Table 14: SBERT scores demonstrating graceful degradation against 21 transforms on COCO-Val2017 images with PixmoCap sentences.

| Transform | LatentSeal | VideoSeal+LLMZip |
|---|---|---|
| Identity | 0.981 | 0.958 |
| HFlip | 0.980 | 0.951 |
| Rot 5 | 0.961 | 0.904 |
| Rot 10 | 0.901 | 0.806 |
| Rot 90 | 0.961 | 0.921 |
| Crop 90 | 0.934 | 0.853 |
| GBlur 5 | **0.623** | **0.305** |
| Sat 0.5 | 0.981 | 0.957 |
| Sat 1.5 | 0.977 | 0.953 |
| Cont 0.5 | 0.975 | 0.948 |
| Cont 1.5 | 0.893 | 0.778 |
| Bright 0.5 | 0.970 | 0.939 |
| Bright 1.5 | 0.887 | 0.771 |
| Hue 0.1 | 0.948 | 0.859 |
| Hue -0.1 | 0.946 | 0.855 |
| Jpeg 80 | 0.923 | 0.888 |
| Jpeg 75 | 0.895 | 0.825 |
| Jpeg 70 | 0.865 | 0.758 |
| Jpeg 60 | 0.768 | 0.603 |
| Jpeg 50 | **0.659** | **0.398** |
| Jpeg 40 | **0.531** | **0.193** |

## C.1 SAMPLING UNIFORMLY RANDOM ROTATIONS ON $\mathbb{S}^{255}$

To perform uniform random rotations on the surface of a 255-dimensional hypersphere ($S^{255}$), we require random rotation matrices drawn from a uniform distribution over the Special Orthogonal group SO(256). A distribution is considered uniform on a compact group if it corresponds to the Haar measure, which ensures that the probability of the matrix falling into any given region of the group is proportional to the volume of that region.

A standard and numerically stable method for generating such matrices is based on the QR decomposition of a matrix of random Gaussian variables. As detailed by Mezzadri [1], a crucial correction step is required to ensure the resulting orthogonal matrix is truly Haar-distributed. The algorithm is described below.

Let $N = 256$. The procedure to generate a single random rotation matrix $O \in$ SO(256) is:

1. **Generate a random matrix from the Ginibre ensemble.** Construct an $N \times N$ matrix $A$, where each element $A_{ij}$ is an independent and identically distributed (i.i.d.) random variable sampled from the standard normal distribution $\mathcal{N}(0, 1)$.

$$A_{ij} \sim \mathcal{N}(0, 1) \quad \text{for } i, j = 1, \dots, N. \tag{8}$$

2. **Perform the QR decomposition.** Factorize the matrix $A$ into the product of an orthogonal matrix $Q$ and an upper-triangular matrix $R$.

$$A = QR \tag{9}$$

where $Q$ is an orthogonal matrix ($Q^T Q = I_N$) and $R$ is an upper-triangular matrix. Most standard linear algebra libraries provide a numerically stable implementation of this decomposition.

3. **Apply the Haar measure correction.** The QR decomposition is not unique. To ensure a uniform (Haar) distribution for the orthogonal matrix, the signs of the diagonal elements of $R$ must be normalized. We construct a diagonal matrix $D$ for this purpose:

$$D_{ii} = \text{sgn}(R_{ii}) = \begin{cases} 1 & \text{if } R_{ii} \geq 0 \\ -1 & \text{if } R_{ii} < 0 \end{cases} \quad \text{and} \quad D_{ij} = 0 \text{ for } i \neq j. \tag{10}$$

Table 15: Watermark model evaluation. We report mean cosine similarity (± standard deviation) between original random and extracted watermarks over 2,000 images from the COCO 2017 validation set. Our model is tested against various augmentations at three target PSNR values (41–43 dB).

| Transform | 41 dB | 42 dB | 43 dB |
|---|---|---|---|
| *Weak* | | | |
| Identity | $0.987 \pm .012$ | $0.984 \pm .018$ | $0.980 \pm .023$ |
| Saturation 0.5 | $0.987 \pm .013$ | $0.984 \pm .018$ | $0.980 \pm .023$ |
| HorizontalFlip | $0.986 \pm .013$ | $0.983 \pm .018$ | $0.979 \pm .024$ |
| Saturation 1.5 | $0.986 \pm .013$ | $0.983 \pm .019$ | $0.979 \pm .024$ |
| Contrast 0.5 | $0.986 \pm .014$ | $0.982 \pm .021$ | $0.978 \pm .026$ |
| Brightness 0.5 | $0.985 \pm .015$ | $0.981 \pm .022$ | $0.976 \pm .028$ |
| *Moderate* | | | |
| GaussianBlur 5 | $0.983 \pm .017$ | $0.979 \pm .024$ | $0.974 \pm .031$ |
| Rotate 90 | $0.981 \pm .017$ | $0.977 \pm .023$ | $0.972 \pm .030$ |
| Hue 0.1 | $0.976 \pm .020$ | $0.972 \pm .026$ | $0.964 \pm .033$ |
| Hue -0.1 | $0.976 \pm .022$ | $0.972 \pm .029$ | $0.965 \pm .036$ |
| Perspective 0.3 | $0.976 \pm .022$ | $0.972 \pm .028$ | $0.965 \pm .036$ |
| JPEG 80 | $0.972 \pm .018$ | $0.965 \pm .026$ | $0.955 \pm .032$ |
| *Strong* | | | |
| Contrast 1.5 | $0.966 \pm .038$ | $0.958 \pm .050$ | $0.952 \pm .053$ |
| Brightness 1.5 | $0.960 \pm .055$ | $0.956 \pm .054$ | $0.947 \pm .066$ |
| JPEG 70 | $0.961 \pm .022$ | $0.949 \pm .030$ | $0.935 \pm .038$ |
| Rotate 10 | $0.949 \pm .043$ | $0.938 \pm .055$ | $0.924 \pm .070$ |
| Perspective 0.5 | $0.942 \pm .053$ | $0.933 \pm .061$ | $0.920 \pm .073$ |
| JPEG 60 | $0.944 \pm .026$ | $0.928 \pm .035$ | $0.909 \pm .043$ |
| JPEG 50 | $0.924 \pm .030$ | $0.903 \pm .042$ | $0.877 \pm .048$ |
| GaussianBlur 17 | $0.913 \pm .078$ | $0.898 \pm .091$ | $0.881 \pm .104$ |
| JPEG 40 | $0.889 \pm .035$ | $0.862 \pm .046$ | $0.827 \pm .052$ |

Table 16: **Landscape of robust image watermarking methods.** Payload capacity is the number of bits embedded per image (RAW is a zero-bit detector that only tests for watermark presence). Only *VideoSeal* openly ships a 256-bit robust and highly imperceptible model. MBRS reaches high payloads but relies on heavy distortion (36 dB) and is not robust beyond JPEG compression, which is why we do not use it as a baseline.

| Watermarking method | Payload capacity (bits) | PSNR (dB) |
|---|---|---|
| LatentSeal (Ours) | $>$**256** | 40–44 |
| VideoSeal Fernandez et al. (2024) | **256** | 40–44 |
| MBRS Jia et al. (2021) | **64–625** | 36 |
| TrustMark Bui et al. (2023) | **100** | 40–44 |
| StegaStamp Tancik et al. (2020) | **32–64** | 42 |
| CIN Ma et al. (2022) | **30** | 36–39 |
| WAM Sander et al. (2025) | **32** /region | 40 |
| RivaGAN Zhang et al. (2019) | **32** | 40 |
| DwtTDcT Al-Haj (2007) | **32** | 40 |
| StableSignature Fernandez et al. (2023) | **48 (model signature)** | 30 |
| RAW Xian et al. (2024) | **0 (presence detection only)** | 27 |

The probability of any $R_{ii}$ being exactly zero is negligible for a continuous distribution. The resulting Haar-distributed orthogonal matrix $Q'$ is then computed as:

$$Q' = QD \tag{11}$$

This procedure ensures that the resulting upper-triangular matrix $R' = D^{-1}R$ has only positive diagonal elements, which makes the decomposition unique and induces the correct Haar measure on $Q'$ [1].

4. **Ensure the matrix is in SO(N) (a pure rotation).** The matrix $Q'$ is guaranteed to be in the orthogonal group O(N), meaning its determinant is either $+1$ or $-1$. The Special Orthogonal group SO(N) contains only matrices with determinant $+1$ (pure rotations).

We thus calculate the determinant of $Q'$. If $\det(Q') = -1$, we convert the matrix into a pure rotation by flipping the sign of any single column. For instance, we can multiply the first column by $-1$:

$$O = \begin{cases} Q' & \text{if } \det(Q') = 1 \\ Q' \cdot \text{diag}(-1, 1, \ldots, 1) & \text{if } \det(Q') = -1 \end{cases} \tag{12}$$

The resulting matrix $O$ is a uniformly distributed random element of SO(256).

This algorithm ensures the security of our watermark, as we can seed the randomness and use this seed as the secret key.

### C.2 IMPACT ON PERFORMANCES

To better understand the impact on performances of this security system, we perform a benchmark on different settings. We benchmark four generators of uniformly distributed random rotations on $SO(d)$:

(i) **np_orig**: Gaussian $\to$ QR $\to$ diagonal sign-fix via $Q \text{diag}(\text{sign}(\text{diag} R))$; then flip one column if $\det(Q) < 0$,

(ii) **np_fast**: identical law, but applies the sign-fix by in-place column scaling of $Q$ to avoid the explicit diag matrix and matmul,

(iii) **torch_batched**: same construction using batched QR on GPU,

(iv) **torch_keyed**: GPU, but each batch element uses its own deterministic RNG stream derived from an integer secret key; same Gaussian$\to$QR pipeline.

All four are Haar on $O(d)$ after the QR sign correction and then restricted to $SO(d)$ by one column flip; only the implementation details differ. We test different keys per batch element because real systems need per-sample determinism and independence without leaking correlation across items. Results are given in Table 17.

**Takeaways** We found that CPU `np_fast` trims a few percent vs. `np_orig` by removing the diag matmul. GPU wins decisively from $d \geq 128$, and batching amortizes kernel overhead further. The per-item keyed variant is a touch slower than fully batched because it must spin a generator per element, but it preserves the same distribution while giving per-sample reproducibility and isolation. Under our setting, a realistic estimate to secure a 256 dimensional latent is at most 1.6 ms by using a T4 Colab GPU.

## D DISCUSSION ABOUT RESULTS

### D.1 DATA CONTAMINATION

We discuss the results in Figure 4. We compare to VideoSeal by leveraging LLMZip with opt-125m to stay in the same weights size range as our text autoencoder. Note that opt-125m is trained on samples from Wikipedia, which might entails test-set leakage for Wikitext. We still think Wikitext is valuable, but strongly urges the reader to take the VideoSeal+LLMZip results with a grain of salt. PixMo-Cap is more recent than OPT models, and is a dataset with new data, not a collection of previous ones, thus we can discard suspicion of test data contamination.

Table 17: Median per-matrix generation time after warm-up. torch_batched benefits from GPU and batching; torch_keyed adds per-item generator setup for deterministic, secret-key–conditioned streams. All methods are Haar-correct on $SO(d)$.

| Dim | Batch | np_orig | np_fast | torch_batched | torch_keyed |
|---|---|---|---|---|---|
| | | | Median time per matrix (ms) | | |
| 32 | 1 | 0.315 | 0.294 | 0.551 | 0.592 |
| 32 | 4 | 0.170 | 0.161 | 0.290 | 0.478 |
| 32 | 16 | 0.160 | 0.155 | 0.119 | 0.470 |
| 32 | 64 | 0.162 | 0.160 | 0.057 | 0.474 |
| 32 | 256 | 0.177 | 0.165 | 0.039 | 0.462 |
| 64 | 1 | 0.467 | 0.486 | 0.552 | 0.633 |
| 64 | 4 | 0.468 | 0.485 | 0.527 | 0.593 |
| 64 | 16 | 0.456 | 0.464 | 0.181 | 0.560 |
| 64 | 64 | 0.515 | 0.756 | 0.091 | 0.747 |
| 64 | 256 | 0.471 | 0.478 | 0.061 | 0.547 |
| 128 | 1 | 2.835 | 2.626 | 1.025 | 1.200 |
| 128 | 4 | 2.714 | 2.630 | 0.722 | 1.228 |
| 128 | 16 | 2.649 | 2.627 | 0.729 | 0.956 |
| 128 | 64 | 2.742 | 2.655 | 0.272 | 0.942 |
| 128 | 256 | 2.742 | 2.632 | 0.133 | 1.089 |
| 256 | 1 | 9.104 | 8.905 | 1.560 | 1.775 |
| 256 | 4 | 9.247 | 8.830 | 1.592 | 1.634 |
| 256 | 16 | 9.097 | 8.726 | 3.588 | 1.591 |
| 256 | 64 | 9.526 | 9.192 | 1.315 | 1.650 |
| 256 | 256 | 9.690 | 9.481 | 0.558 | 1.600 |

## D.2 DATASET DIFFICULTY

In order to better understand our findings, we analyze datasets composition. We expect Wikitext to be more difficult, as it is sourced from Wikipedia in which each page is full of named entities with few to no occurrences. Figure 5 confirms this assumption and showcases the low overlap (27%) between 4-grams from train and test sets. This means our system has to generalize in order to achieve good results on test set. Conversely, we perform the same study on PixMo-Cap and find that our random split is way worse in terms of overlap, meaning that good performances may be due to this overlap between train and test sets. To further probe our system abilities, we use the novelty score introduced in Section 4.4 to curate the 5% of samples with the lowest count of common n-grams between train and test sets. As stated before, we observe a drop of 5 of BLEU4, which is sound and ensures that our system does work.

## D.3 RELATION BETWEEN NUMBER OF TOKENS AND BIT PAYLOAD

In Figure 6, we estimate the bit length associated with a number of tokens using LLMZip on all ours dataset splits. This enables us to target 30 tokens as a fair ground for comparison with the 256 bits capacity of VideoSeal.

To better emphasizes the current limitations of a 256 bits system, Fig. 7 depicts the rate of samples going above this threshold for a $N$ token long message in Wikitext. We estimate bit length using LLMZip over the 4,000 sentences. So for 30 tokens, about $16\%$ of sentences are above the 256 bits threshold and will be partially transmitted due to limited capacity.

Note that the relation between the latent space dimension and bit payload is not straightforward, and cannot be easily derived. In our case, this mainly relies on the capacity of the text encoder to compress information from a given number of tokens in a given latent space dimension.

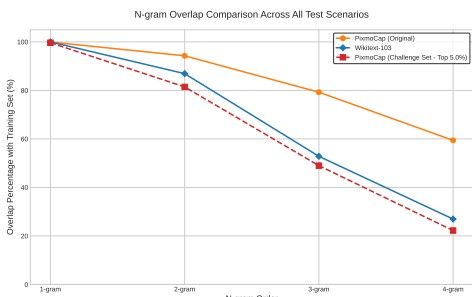

Figure 5: We study train/test ngram overlap as an heuristic to determine how much the model is able to generalize on new data. We show that PixMo-Cap test set is quite repetitive and thus design a metric to optimize the hardness of the test set, keeping only the top hardest 5%.

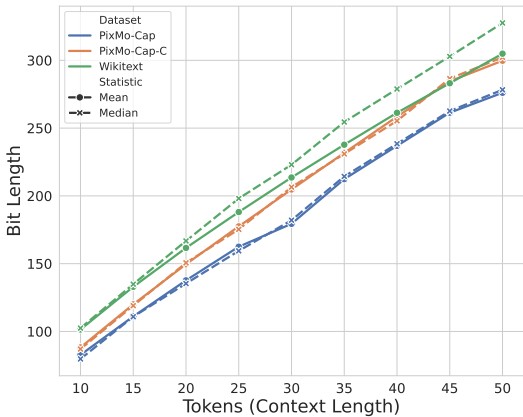

Figure 6: Relationship between the mean number of tokens and the compressed bit length using LLMZip with OPT-125M, computed on Wikitext, PixmoCap and PixmoCap-C (for the harder split).

## E  IMAGE EDITING

We detail the influence on perceptibility, through the PSNR metric, of each image editing model in Table 18. Samples can be found in Fig. 9. We show in Figure 8 state-of-the-art performances in comparison with the VideoSeal baseline.

Table 18: PSNR statistics for HiDream and ip2p image-editing models.

| Statistic | HiDream (dB) | ip2p (dB) |
|---|---|---|
| Mean | 15.6372 | 19.8388 |
| 1st quartile | 10.2666 | 13.7499 |
| Median | 14.8781 | 19.7041 |
| 3rd quartile | 20.0194 | 25.3874 |

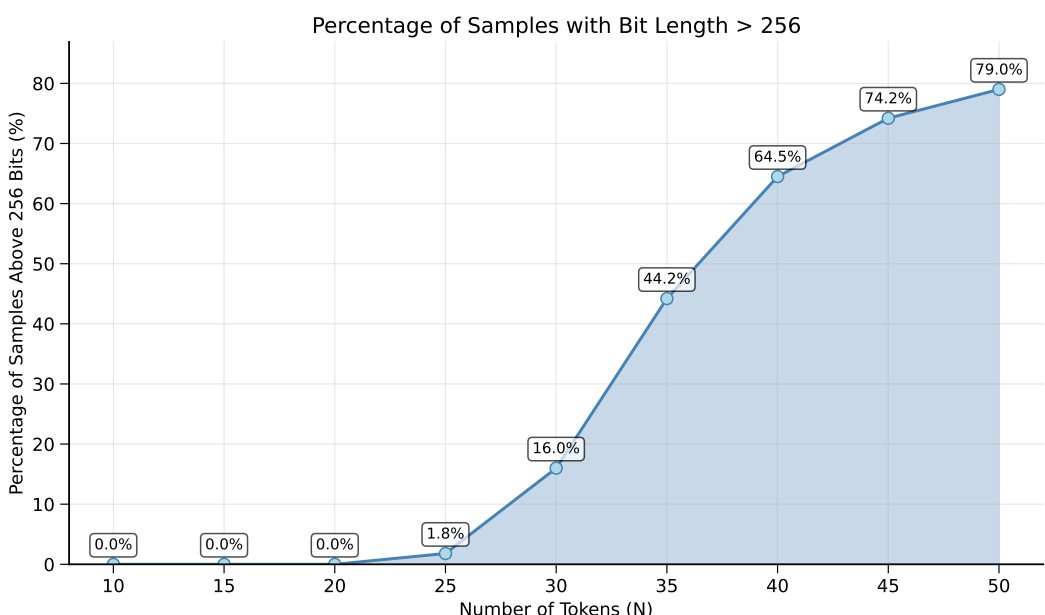

Figure 7: On Wikitext, we compute the percentage of samples above the maximum capacity of Videoseal for better emphasis of existing limitations.

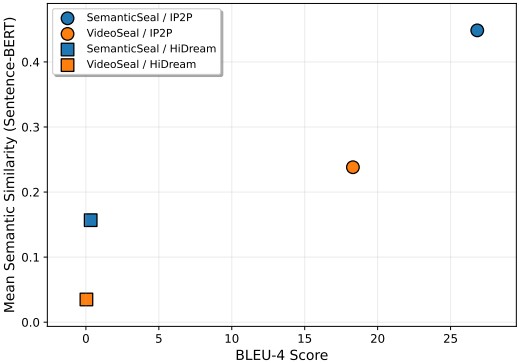

Figure 8: BLEU4 and SBERT similarity between original and retrieved watermarks after image editing on cover. Our method obtains SOTA results in terms of robustness.

# F  QUALITATIVE EXAMPLES

## F.1  WATERMARKED DATA SAMPLES

We provide some instances in which LatentSeal was used in Figure 10. Depending on the application, the message may be aligned with the cover image, but is not required to. Textual messages can be sourced from human operators aiming to transmit a specific message (aligned or not with the image content), or from an image captioning model.

## F.2  TEXT DATA SAMPLES

In this section, we exhibit representative samples for PixMo-Cap et Wikitext dataset.

**PixMo-Cap**

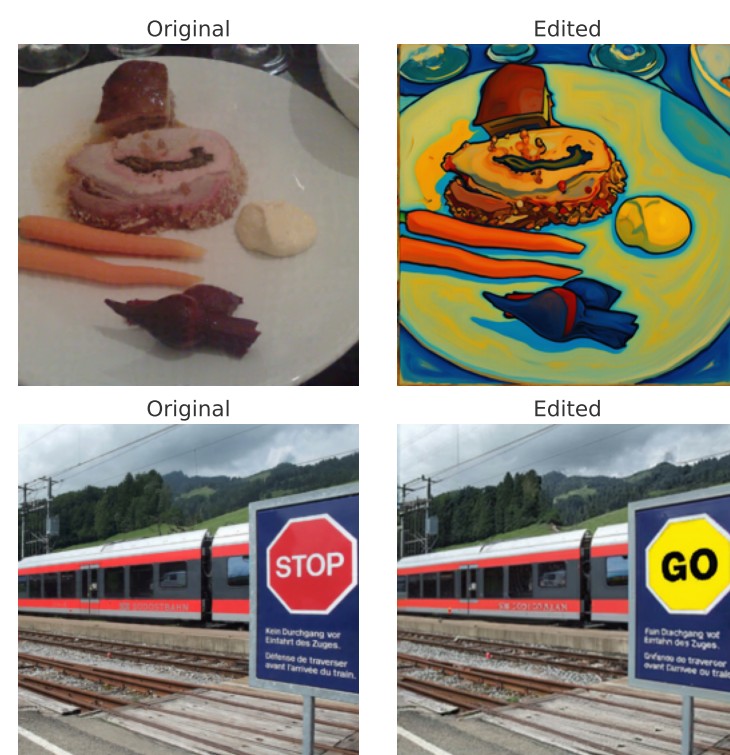

Figure 9: Samples of images edited with HiDream

- "Close-up photograph of a gourmet grilled cheese sandwich that has been artistically sliced in half. Each half reveals a gooey, white cheese center with an enticing stringy, melted cheese bridge connecting them. The sandwich features double"

- "The image depicts a bustling street scene set in a populated area, prominently featuring historical and military elements associated with Checkpoint Charlie. On the left side of the street stands a four-story building labeled "Checkpoint Charlie Wall Museum," "

- "This informative and somewhat humorous poster centers around an image of two tortoises, one seemingly mounting the other for reproductive purposes. The composition has the tortoises positioned centrally, with text above and below them."

- "This photograph, taken in a hurried or unsteady moment, offers a blurry glimpse into a cozy kitchen corner. The focus is on the intersection of counters and cabinets, where white cabinetry adorns both the upper and lower sections."

- "The front page of the clothing website "Apparelix" is designed with a clean, white background. Located in the upper left corner is "USD" in gray text, accompanied by an arrow that likely allows users to change the pricing"

**Wikitext**

- " Robert Boulter is an English film , television and theatre actor . He had a guest @-@ starring role on the television series The Bill in 2000 . This was followed by a starring role in the play Herons written by Simon Stephens , which was performed in 2001 at the Royal Court Theatre . He had a guest role in the television series Judge John Deed in 2002 . In 2004 Boulter landed a role as " Craig " in the episode " Teddy 's Story " of the television series The Long Firm ; he starred alongside actors Mark Strong and Derek Jacobi . He was cast in the 2005 theatre productions of the Philip"

- " In 756 , Emperor Xuanzong was forced to flee the capital and abdicate . Du Fu , who had been away from the city , took his family to a place of safety and attempted to join the

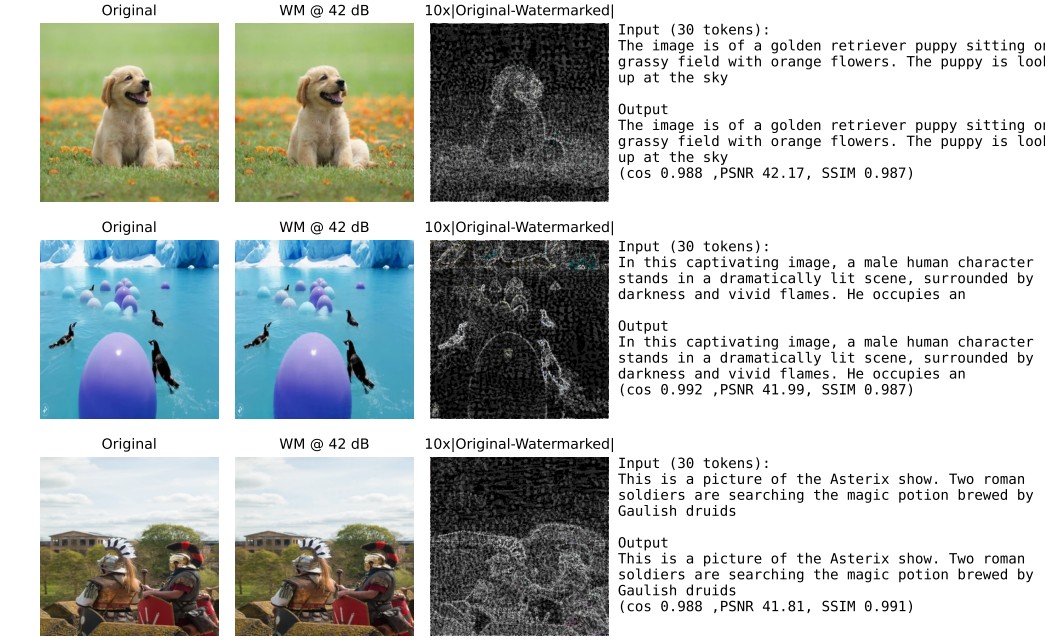

Figure 10: Qualitative samples of LatentSeal. We report the original text and the decoded one, as well as cosine similarity and perceptual metrics. First row input is from Florence-2-base, a captioning model. Second is from PixMo-Cap dataset. And last input is human made, thus out of distribution.

court of the new emperor ( ¡unk¿ ) , but he was captured by the rebels and taken to Chang 'an . In the autumn , his youngest son , Du ¡unk¿ ( Baby Bear ) , was born . Around this time Du Fu is thought to have contracted malaria . "

• " The naming of these first specimens was disputed , however . Leopold Fitzinger named the animal ¡unk¿ in 1837 . In 1841 , English palaeontologist Richard Owen referred to the genus as Labyrinthodon to describe its highly folded or labyrinthine teeth . Owen thought the name Mastodonsaurus " ought not to be retained , because it recalls unavoidably the idea of the mammalian genus Mastodon , or else a ¡unk¿ form of the tooth ... and because the second element of the word , saurus , indicates a false affinity , the remains belonging , not to the Saurian , but to the ¡unk¿"

