# OpenReview forum: "Fast, Secure, And High-Capacity Image Watermarking With Text Autoencoded Text Vectors"
_ICLR.cc/2026/Conference — Submitted to ICLR 2026_

### Official Review · Reviewer_era7 · 2025-10-16

**Soundness:** 2
**Presentation:** 2
**Contribution:** 2
**Rating:** 4
**Confidence:** 4

**Summary:**

This paper presents a novel image watermarking method called LatentSeal, which shifts the watermarking paradigm from the traditional "bitstream" approach to a "semantic communication" model. It introduces a key-based latent-space rotation encryption mechanism, achieving a unified balance of high capacity, robustness, security, and semantic interpretability.

**Strengths:**

1. The method embeds meaningful textual information within the watermark, enhancing its interpretability.
2. The effectiveness and robustness of the watermarking scheme have been validated across multiple datasets and attack scenarios.

**Weaknesses:**

1. The structure of the proposed model is relatively simple and seems to be a combination of existing works.
2. The experimental metrics only consider BLEU-4 and EM, without incorporating measures related to watermark strength.

**Questions:**

1. The text encoder and watermark embedding model used in this work are based on existing approaches. Where does the core innovation of this paper lie?
2. The watermark model and text autoencoder in this work require staged training. Could an end-to-end training approach be considered instead?
3. The paper claims to "break the 256-bit payload limit," but this is achieved by compressing the text into a continuous vector space, which seems to differ from the traditional definition of "bits" in information theory. Could you clarify this?

---

> ### Author Response · Authors · 2025-11-13
>
> We thank the reviewer for acknowledging that our approach embeds semantically meaningful text in the watermark, which improves interpretability, and that the effectiveness and robustness of the scheme are demonstrated across multiple datasets and a broad range of attack scenarios.
>
> > 1. The experimental metrics only consider BLEU-4 and EM, without incorporating measures related to watermark strength.
>
> We interpret "watermark strength" as the magnitude of the perturbation applied to the original image, i.e., its imperceptibility. In our experiments, a measure related to this quantity is explicitly reported via the PSNR between the original and watermarked images.
>
> In Section 4.1 (*Watermarking model robustness*), we state that all robustness results are obtained at a fixed watermark strength of approximately $42\,\mathrm{dB}$ PSNR: *“We watermark at 42 dB in our experiments as we require high-standard imperceptibility.”* This is a standard measure linked to watermark strength in the watermarking literature, and most recent works report results around $40\,\mathrm{dB}$. Operating at $42\,\mathrm{dB}$ therefore corresponds to a stricter imperceptibility regime. BLEU-4 and EM are then used on top of this fixed-strength setting to evaluate the semantic fidelity of the recovered messages.
>
> We will revise the paper to make this link more explicit by:
> (i) clearly referring to PSNR as the *watermark strength / imperceptibility* metric in Section 4.1, and
> (ii) explicitly pointing from Section 4.1 to Appendix Table 11, which studies performance as a function of PSNR in the range 41–43 dB, and to Appendix Table 12, which summarizes typical PSNR ranges used in prior work.
>
> Concretely, watermark strength is controlled in our implementation by a scalar factor $\alpha$ that multiplies the watermark residual before it is added to the image. This allows us to match the “watermarking budget” when comparing to VideoSeal: for all main experiments we fix $\alpha = 0.20$, which corresponds to approximately $42\,\mathrm{dB}$ PSNR.
>
> To further address the reviewer’s request for metrics related to watermark strength, we will add complementary perceptibility metrics to already reported PSNR with SSIM and LPIPS and report how they vary with $\alpha$, together with the cosine similarity between the original and recovered vectors. Those results are performed on 2022 images from the test set of Emu Edit with 8 variations of watermark strength and corresponding metrics on both perceptibility and detection. We plan to add the following table to main text with previous explanations:
>
> | Power | **Perceptibility Metrics** |        |         | **Cosine Similarity** |        |         |       |       |
> |:-----:|:--------------------------:|:------:|:-------:|:---------------------:|:------:|:-------:|:-----:|:-----:|
> |   $\alpha$   | PSNR↑                      | SSIM↑  | LPIPS↓  | Clean                 | JPEG-50| JPEG-30 | Noise | Crop  |
> | 0.05  | 49.2                       | 0.9982 | 0.0008  | 0.7581                | 0.5537 | 0.4068  | 0.6152| 0.6132|
> | 0.10  | 47.14                      | 0.9967 | 0.0017  | 0.9250                | 0.8193 | 0.6910  | 0.8567| 0.8658|
> | 0.15  | 44.43                      | 0.9940 | 0.0031  | 0.9697                | 0.9235 | 0.8507  | 0.9394| 0.9452|
> | **0.20** | **42.20**               | **0.9897** | **0.0055** | **0.9837**   | **0.9601** | **0.9200** | **0.9677** | **0.9693** |
> | 0.25  | 40.41                      | 0.9844 | 0.0088  | 0.9877                | 0.9726 | 0.9465  | 0.9774| 0.9774|
> | 0.30  | 38.92                      | 0.9780 | 0.0134  | 0.9886                | 0.9778 | 0.9591  | 0.9815| 0.9798|
> | 0.35  | 37.66                      | 0.9713 | 0.0186  | 0.9881                | 0.9790 | 0.9635  | 0.9819| 0.9794|
> | 0.40  | 36.55                      | 0.9624 | 0.0263  | 0.9868                | 0.9800 | 0.9675  | 0.9817| 0.9778|
>
> This table shows that $\alpha = 0.20$ achieves a favorable trade-off: PSNR ≈ 42 dB with SSIM ≥ 0.989 and low LPIPS, while preserving high cosine similarity of the recovered message even under common distortions.
>
> We hope this address concerns on the lack of metric related to watermark strength.

---

> ### Author Response · Authors · 2025-11-13
>
> > 2. The structure of the proposed model is relatively simple and seems to be a combination of existing works.
> The text encoder and watermark embedding model used in this work are based on existing approaches. Where does the core innovation of this paper lie?
>
> We agree that LatentSeal is intentionally built from well-established components (ModernBERT, an adapted VideoSeal backbone, and standard cryptographic primitives), and this design choice is deliberate: it facilitates modularity, interpretability, and practical adoption. The novelty of our work lies in how these components are repurposed and coupled to realize a new watermarking setting: semantic, continuous, key-conditioned watermarking with calibrated confidence, rather than bit-level payloads. In addition, we will release the full implementation (text autoencoder, watermarking model, and training code), which makes this paradigm directly usable and easily extensible for the community.
>
> Concretely:
>
> -   **Latent watermarking model.** We re-engineer VideoSeal from a
>     256-bit classifier operating on $\{0,1\}^{256}$ into a continuous
>     channel on the unit hypersphere $\mathbb{S}^{255}$. This requires
>     architectural changes and complete retraining with a
>     cosine-similarity objective, since the original model does not
>     support continuous, normalized inputs. The resulting latent
>     watermarker is, to our knowledge, the first robust, publicly
>     available model designed to embed high-dimensional continuous
>     unit-norm vectors into images (Appendix B).
>
> -   **Task-specific text autoencoder.** On the text side, we leverage
>     ModernBERT as the encoder, but learn the projection from the \[CLS\]
>     token into the latent space and the entire decoder from scratch. The
>     decoder is conditioned on the watermark via a cross-attention
>     mechanism, and we inject noise in the latent space during training.
>     This combination (noise in the bottleneck + watermark-conditioned
>     decoding) is not present in standard text autoencoders and is
>     tailored to make the latent representation robust and compatible
>     with the watermarking channel.
>
> -   **Security and detection layer.** We introduce a key-conditioned
>     orthogonal rotation to provide message confidentiality in the spirit
>     of Kerckhoffs' principle \[1\], and a statistically calibrated
>     confidence score (Eq. (4)) with operational thresholds (Table 4) to
>     decide when a decoded message should be trusted. Recent post-hoc
>     watermarking works typically omit such key-based security and
>     calibrated detection.
>
> As a result, the overall architecture remains structurally simple, but
> the way these components are adapted and combined is new: LatentSeal
> turns watermarking from a bit-centric detection problem into a semantic
> communication channel with security and confidence. Empirically, Table 3
> shows that under this more demanding setting (semantic messages,
> continuous latents), LatentSeal matches or improves over the
> VideoSeal+LLMZip baseline at equal payload on multiple datasets and
> perturbations, and additionally enables reliable decoding for *higher*
> payloads (e.g., WikiText messages $>$ 256 bits where VideoSeal+LLMZip
> collapses to 0% EM).
>
> \[1\] Cox I J, Miller M L, Bloom J A, et al. Digital watermarking.
> Morgan Kaufmann Publishers, 2008.

---

> ### Author Response · Authors · 2025-11-13
>
> > 3. Why not end-to-end training for watermarker and autoencoder?
>
> We agree that, in principle, end-to-end training is an appealing idea.
> In our setting, however, it is neither necessary nor clearly beneficial.
>
> First, a key property of our watermarking model is that it is trained to
> hide *random* vectors drawn from $\mathbb{S}^{255}$, sampled via a pseudo-random generator $G_k$ conditioned on the secret key $k$. The watermarker learns to embed and extract uniformly distributed latents on the unit sphere, independent of any particular upstream encoder. Under the null hypothesis $\mathcal{H}_0$ (extracting with a random key), the extracted vectors are uniformly distributed on $\mathbb{S}^{255}$ due to the induced random rotation. Therefore there is no structure to learn from the distribution of the latent vector after rotation, that the watermarking system could benefit from. In other words, the watermarking model is not biased toward any specific subset of latent vectors.
>
> Admitting we accept dropping the security constraint, coupling the watermarker and the text autoencoder in an end-to-end training loop would indeed specialize the watermarker to the latent distribution of this specific encoder. However, it would also prevent the use of a common system to encode other continuous embeddings, e.g. from other modalities than text. Our current setting achieves an average cosine similarity of about $0.98$ between embedded and extracted vectors on clean images, indicating that the watermarking channel is already highly accurate without additional specialization.
>
> Second, from an optimization perspective, both the watermarking model
> and the text autoencoder are non-trivial to train on their own. Jointly
> tuning them would significantly enlarge the hyperparameter space and
> introduce additional sources of instability (e.g., balancing
> reconstruction losses across two coupled networks). Given the strong
> performance of the decoupled approach, we expect any gains from
> end-to-end training to be marginal relative to the added complexity and
> risk of non-convergence.
>
> For these reasons, we favor a modular training strategy: (i) it offers
> better generalization to arbitrary continuous latents on
> $\mathbb{S}^{255}$, (ii) it avoids unnecessary optimization complexity,
> and (iii) it makes the components reusable, since the watermarker can be
> paired with other encoders without retraining the full system. We will
> clarify these design choices in the revised version.

---

> ### Author Response · Authors · 2025-11-13
>
> > 4. The paper claims to 'break the 256-bit payload limit,' but this is achieved by compressing the text into a continuous vector space, which seems to differ from the traditional definition of 'bits' in information theory. Could you clarify this?
>
> Thank you for raising this point. We agree that our notion of "payload"
> differs from the traditional, information-theoretic definition of a
> fixed number of discrete bits. This is intentional: our goal is to
> reframe watermarking as *semantic communication* rather than as a raw
> bit pipe.
>
> In the paper, the phrase "breaking the 256-bit limit" is meant in a
> *practical* sense, in direct comparison to state-of-the-art multibit
> methods such as VideoSeal, which embed at most 256 discrete bits per
> image. To make this comparison fair, we estimate the information content
> of the text messages we embed using LLMZip, as described in lines
> 302--304. In Appendix D.3 (Figures 6 and 7), we show that standard
> 30-token messages from a complex dataset such as WikiText often require
> significantly *more* than 256 bits to be losslessly compressed by a
> strong compressor like LLMZip.
>
> In such cases, a strictly 256-bit watermarking system is forced to
> truncate the message: it cannot faithfully transmit the full text, even
> if its 256 bits are used optimally. By contrast, LatentSeal maps the
> entire sentence into a 256-dimensional continuous unit-norm vector and
> trains the decoder to reconstruct the full text from that latent.
> Empirically, we observe that LatentSeal can recover messages whose
> LLMZip-compressed size exceeds 256 bits, whereas the VideoSeal+LLMZip
> baseline cannot, due to truncation (see Table 3 and Appendix D.3).
>
> In other words, our 256-dimensional latent space is not constrained to encode the information to an arbitrarily fixed number ($2^{256}$) of codewords. Rather, it is a continuous semantic representation that is robust to noise introduced by the watermarking system and the alteration of the cover image. Under practical scenarii where this alteration is generally limited and unintentional, it enables to reliably convey sentences whose lossless compression length is often above 256 bits. We will clarify this in the main text by explicitly stating that our claim of "breaking the 256-bit limit" is a practical, baseline-relative statement.

---

> ### Author Response · Authors · 2025-11-25
>
> Dear Reviewer,
>
> Thank you again for your constructive comments. We hope the clarifications provided in our response and in the revised version adequately addressed your questions. If any part of our submission would benefit from additional material, we are happy to elaborate !

---

> > ### Comment · Reviewer_era7 · 2025-11-26
> >
> > Thank you for the authors' clarification. However, I still have reservations regarding the novelty of the paper, and therefore I will maintain my original score.

---

> > > ### Author Response · Authors · 2025-11-26
> > >
> > > Dear Reviewer,
> > >
> > > Thank you for your answer. We respectfully disagree that novelty is lacking, and would like to better understand your concern.
> > >
> > > Last years prior work in multibits watermarking has focused on increasing bit capacity while maintaining robustness. Our work proposes a different path: semantic watermarking, where the payload is interpretable natural language rather than meaningless bits. This enables applications like tamper explanation, not just detecting manipulation, but revealing what changed. This only is novel.
> > >
> > > Making this work required other novelties:
> > >
> > > - Adapting VideoSeal to embed continuous latents on $\mathbb{S}^{255}$ (new objective, full retraining)
> > > - A noise-robust text autoencoder designed for channel transmission
> > > - Calibrated confidence scoring for deployment decisions
> > > - Post-hoc key-based security
> > >
> > >
> > >
> > > The empirical results validate the paradigm: 51.6% EM on >256-bit messages where VideoSeal+LLMZip gets 0%, with 121× faster decoding and security (which is not optional).
> > > None of these components is present in the recent watermarking literature and we commit to open-source all of them.
> > >
> > > We believe this setting is new. If you are aware of prior work achieving such watermarking with comparable speed, security, capacity, robustness, and interpretability we would genuinely appreciate the reference.
> > >
> > >
> > > For future directions, our latent watermarking model embeds any unit-norm vector on $\mathbb{S}^{255}$, not just text representations, thanks to the independant trainings. This opens possibilities for cross-modal watermarking, embedding audio, image, or other feature vectors into images, which multibits methods **cannot** naturally support.
> > >
> > > Could you help us understand which specific aspect you find insufficiently novel with respect to previous work?

---

> > > > ### Comment · Reviewer_era7 · 2025-11-26
> > > >
> > > > Is embedding meaningful textual vectors truly the core novelty of this work? In my view, the semantic interpretation of bits can be defined in an arbitrary manner. For example, an $n$-bit message can be associated with $2^n$ possible meanings. For this reason, I am not fully convinced that this aspect represents a fundamental challenge that needs to be addressed. In addition, since the watermark embedding and extraction procedures largely rely on existing techniques, I feel that the contribution may fall somewhat short of the bar typically expected for acceptance at ICLR.

---

> > > > > ### Author Response · Authors · 2025-11-27
> > > > >
> > > > > Thank you for your remark, which really helps us understand where you're coming from.
> > > > >
> > > > > You are right that $n$ bits can map to $2^n$ meanings via a table. 256 bits allows for $10^{77}$ entries, which combined with a shared table of meanings would allow to communicate a lot of pre-established messages. However, mapping a more generic sentence to a specific ID would be intractable, and searching for the closest meaning in the shared table would still be challenging. In a similar way, images for example are not encoded by assigning them an ID in a gigantic database, but rather encoded with codecs such as JPEG that produce much larger bitstreams while generalizing to describe new and unseen content more easily. This is why we propose to compare against LLMZip as a strong baseline for text compression.
> > > > >
> > > > > LLMZip uses LLM probabilities for lossless arithmetic coding, which makes it adaptive to the text domain the LLM was trained for. Yet even so, ~16% of 30-token sentences exceed 256 bits and get truncated (Fig.7). Also, using a LLM for high-quality probabilities prediction makes the system quite slow (Fig.3).
> > > > >
> > > > > As an alternative, our autoencoder does not produce a variable-length bit sequence. Rather, it learns to map sentences to a continuous 256-D space, trained to be robust to noise. Any sentence may be encoded to this space, and even if the channel is noisy, will be reconstructed to a sentence semantically close rather than a hard failure, provided the sentence is in-domain. Graceful degradation is a contribution of our work, obtained from working in continuous space rather than fixing an arbitrary number of message bits in advance.
> > > > >
> > > > > The SBERT table below show the graceful degradation (measured with SBERT) against VideoSeal+LLMZip on 1000 images with 256 bits messages from PixmoCap, declined on a set of 21 transforms. We manage to always achieve a better SBERT score, and under heavy attack (GaussianBlur, Jpeg50&40) we observe huge improvement.
> > > > >
> > > > >
> > > > > | Transform   | LatentSeal | VideoSeal+LLMZip |
> > > > > |-------------|------------|-----------|
> > > > > | Identity    | 0.981      | 0.958     |
> > > > > | HFlip       | 0.980      | 0.951     |
> > > > > | Rot 5       | 0.961      | 0.904     |
> > > > > | Rot 10      | 0.901      | 0.806     |
> > > > > | Rot 90      | 0.961      | 0.921     |
> > > > > | Crop 90     | 0.934      | 0.853     |
> > > > > | GBlur 5     | **0.623**      | **0.305**     |
> > > > > | Sat 0.5     | 0.981      | 0.957     |
> > > > > | Sat 1.5     | 0.977      | 0.953     |
> > > > > | Cont 0.5    | 0.975      | 0.948     |
> > > > > | Cont 1.5    | 0.893      | 0.778     |
> > > > > | Bright 0.5  | 0.970      | 0.939     |
> > > > > | Bright 1.5  | 0.887      | 0.771     |
> > > > > | Hue 0.1     | 0.948      | 0.859     |
> > > > > | Hue -0.1    | 0.946      | 0.855     |
> > > > > | Jpeg 80     | 0.923      | 0.888     |
> > > > > | Jpeg 75     | 0.895      | 0.825     |
> > > > > | Jpeg 70     | 0.865      | 0.758     |
> > > > > | Jpeg 60     | 0.768      | 0.603     |
> > > > > | Jpeg 50     | **0.659**      | **0.398**    |
> > > > > | Jpeg 40     | **0.531**      | **0.193**     |

---

> > > > > ### Author Response · Authors · 2025-11-27
> > > > >
> > > > > Another important contribution related to the watermark representation choice is the introduction of security that becomes almost trivial. Because our latents live on $S^{255}$, we can apply a secret key-conditioned rotation matrix from $SO(256)$. It's invertible, preserves the norm, and is very fast. Without the key, the extracted latent is just a random point on the sphere, and the decoder outputs a totally unrelated sentence. So we also provide a security system working with our setting, making it suitable for real-world use.
> > > > >
> > > > >
> > > > >
> > > > > So from one design choice and several models, we get: higher capacity, graceful degradation under noise, faster decoding and cheap post-hoc security. These contributions all stem from our representation on $\mathbb{S}^{255}$.
> > > > >
> > > > > We agree the watermarking backbone builds on VideoSeal, as it couldn't embed continuous vectors out of the box, we had to change the loss and retrain it. Even if you may consider this as a relatively small modification, to the best of our knowledge none of the state-of-the-art watermarking systems ever proposed this alternative to get rid of the dependency on an arbitrary fixed message length, and rather compete on its length and bit-error rates. We argue that this fixed-length binary message design is an artificial limitation, and show applications of departing from it in this work.
> > > > > On the other hand, the autoencoder's use in crafting vectors suited for watermarking and for decoding robustly is new and represents a significant advancement for future work as we found no equivalent available in previous work, most likely due to our bottleneck dimension which is very low and our noise added during training. We strongly believe this paves the way to additional research on the autoencoder part to improve the denoising ability of this architecture.
> > > > >
> > > > > The fundamental challenges adressed here are increasing message capacity and overcoming the intrinsic limitation of an arbitrary number of bits, and reintroducing security.
> > > > > So rather than "embedding meaningful textual vectors", we'd frame the core novelty as: "embedding continuous vectors enables meaningful, secure watermarking with graceful degradation and fast decoding." The distinction matters because these properties emerge jointly from the representation choice. And again, it is hard to only focus on this, as we think the autoencoder idea is a strong contribution to future watermarking framework.
> > > > >
> > > > > We think that all these contributions justify the need to explore this research direction in watermarking, and is worth publishing at ICLR.
> > > > >
> > > > > Hopefully this will help clarify what we see as core contributions. We will be happy to continue discussing the matter, and appreciate you taking the time to give us this feedback.

---

### Official Review · Reviewer_AGKw · 2025-10-28

**Soundness:** 3
**Presentation:** 2
**Contribution:** 2
**Rating:** 2
**Confidence:** 4

**Summary:**

This paper proposes LatentSeal, a watermarking method that embeds a message sentence into an image. LatentSeal encodes a message into a unit vector, applies a secret rotation, and embeds it into the image. The message is later decoded by extracting the embedded vector. Building upon VideoSeal, the embedder of LatentSeal is empirically shown to be robust against common image edits.

**Strengths:**

1. The motivation and overall method are clearly presented and easy to follow.

2. Using semantically meaningful vectors as watermarks is a natural and appealing idea.

**Weaknesses:**

This paper proposes LatentSeal, a watermarking method that embeds a message sentence into the image. LatentSeal encodes a message into a unit vector, applies a secret rotation, and embeds it to an image. The message is decoded after extracting the embedded vector. Building upon VideoSeal, the embedder of LatentSeal is empirically shown to be robust to common image edits.



**Strengths:**

1. The motivation and main method are clearly explained and easy to follow.
2. Using sematic meaningful vectors as watermark seems to be a natural and appealing idea.




**Concerns:**

1. Comparison to existing work. My understanding is that LatentSeal is autoencoder + VideoSeal. It is helpful to further explain the difference and innovation compared to VideoSeal.
2. Watermark detection. The detection mechanism and its justification remain unclear. Specifically, in Figure 1, $\hat{y}$ denotes the latent vector extracted from an image, and $\hat{y}{\mathrm{rec}} = E \odot D(\hat{y})$, where $E$ and $D$ represent the encoder and decoder, respectively. The authors claim that an image is authentic if $\hat{y}$ is close to $\hat{y}{\mathrm{rec}}$. However, for any in-distribution latent vector $y$, a well-trained autoencoder typically satisfies $E \odot D(y) \approx y$. This suggests that closeness between $\hat{y}$ and $\hat{y}_{\mathrm{rec}}$ alone may not imply authenticity, as the relationship between $\hat{y}$ and the true latent vector $y$ remains unknown. Could the authors clarify how the confidence score in Equation (4) effectively captures this relationship?
3. Secret Rotation. What is the motivation for introducing the secret rotation layer? Please provide an example or real-world scenario where this component is necessary or advantageous for security or robustness.
4. How does the method’s performance change if the latent vector dimensionality is altered (e.g., not fixed at 255)? Some empirical evidence would strengthen the claims of generality.
5. I would suggest add more basline methods such as DwtDct [1], Stable Signature [2], RAW [3], and etc. Comparing with them in terms of the AUC-ROC score helps to understand the cost of embedding meaningful messages.

[1] Cox I J, Miller M L, Bloom J A, et al. Digital watermarking[J]. Morgan Kaufmann Publishers, 2008, 54: 56-59.

[2] Fernandez P, Couairon G, Jégou H, et al. The stable signature: Rooting watermarks in latent diffusion models[C]//Proceedings of the IEEE/CVF International Conference on Computer Vision. 2023: 22466-22477.

[3] Xian X, Wang G, Bi X, et al. Raw: A robust and agile plug-and-play watermark framework for ai-generated images with provable guarantees[J]. Advances in Neural Information Processing Systems, 2024, 37: 132077-132105.

**Questions:**

Please see Weaknesses above.

---

> ### Author Response · Authors · 2025-11-17
>
> We thank the reviewer for the constructive assessment and
> for highlighting that (i) the motivation and main method are clearly
> explained and easy to follow, and (ii) using semantic, meaningful latent
> vectors as watermarks is a natural and appealing idea. The main concerns
> raised focus on clarification of our detection mechanism, the role of
> the secret rotation, the effect of latent dimensionality, the
> relationship to VideoSeal, and the positioning with respect to classical
> and recent baselines.
>
> In summary, we believe these concerns can be addressed without changing
> the core claims of the paper. In the following answers we:
>
> 1.  clarify how LatentSeal differs from and extends VideoSeal beyond a
>     simple "autoencoder + VideoSeal" combination, including the shift
>     from a bitstream to a continuous unit-sphere channel;
>
> 2.  explain the watermark detection mechanism and justify the confidence
>     score in Eq. (4), emphasizing the non-triviality of the relationship
>     between $\hat{y}$ and $\hat{y}_{\text{rec}}$ once the watermarked
>     image has undergone perturbations;
>
> 3.  motivate the secret, key-conditioned rotation with a concrete
>     security/provenance scenario, and make explicit that it targets
>     confidentiality rather than robustness to image distortions;
>
> 4.  report and discuss the effect of varying the text latent
>     dimensionality (128--384) and justify our choice of 256 dimensions
>     based on the empirical sweep provided in the appendix;
>
> 5.  clarify why direct AUROC comparisons with DwtDct, StableSignature,
>     and RAW are not directly meaningful given their very different
>     payload regimes and detection objective,
>     while still including them in the revised version for completeness.

---

> ### Author Response · Authors · 2025-11-17
>
> > 1. Comparison to existing work. My understanding is that LatentSeal is autoencoder + VideoSeal. It is helpful to further explain the difference and innovation compared to VideoSeal.
>
> We agree that our framework integrates two main components: a text
> autoencoder and a watermarking model adapted from VideoSeal. We also
> view this modular design as a strength for interpretability and future
> extensions, and we will release the code for both components upon
> publication. However, the description "autoencoder + VideoSeal"
> overlooks the substantial changes required to make this combination work
> in our semantic-communication setting.
>
> Our key contribution lies in how these components are adapted and
> coupled. On the watermarking side, we do *not* use VideoSeal
> off-the-shelf. Instead, we shift it from a discrete bit-pipe to a
> continuous vector channel:
>
> -   **Different operational domain.** VideoSeal is designed to embed and
>     extract bitstreams in $\{0,1\}^{256}$. In LatentSeal, the
>     watermarker operates on the continuous unit hypersphere
>     $\mathbb{S}^{255}$, embedding 256-dimensional unit-norm vectors.
>     This required architectural modifications and complete retraining,
>     since the original model does not support this normalized,
>     continuous space. We detail the exact changes in Appendix B and
>     release the adapted model.
>
> -   **Different learning objective.** Consequently, the training
>     objective is also changed. We replace the Binary Cross-Entropy loss
>     for bit classification with a **cosine similarity loss**, which is
>     crucial to preserve the geometry of the latent space and to make the
>     decoder's confidence score well-calibrated.
>
> -   **New public artifact.** The resulting watermarker is, to our
>     knowledge, the first robust, publicly available model specifically
>     designed to embed high-dimensional, continuous unit-norm vectors
>     into images, enabling other applications beyond text (image descriptors).
>
> On the autoencoder side, we train a text autoencoder specifically
> tailored to take advantage of watermarking. The decoder uses a conditioning
> mechanism compatible with the watermark that leverages cross-attention memory, and we inject noise
> during training to improve robustness of the latent representation,
> which is not standard in generic text autoencoders.
>
> To make the distinction with the original VideoSeal more obvious, Table 3
> shows that LatentSeal globally matches or improves the EM/BLEU scores on
> PixMo-Cap and COCO-2017, and,
> crucially, maintains non-trivial exact match on WikiText for messages
> longer than 256 bits, whereas VideoSeal+LLMZip collapses to 0% EM due to
> truncation. This highlights that LatentSeal operates in a different
> regime: a continuous latent channel designed for semantic messages, with
> state-of-the-art performance under the same or more demanding
> conditions.
>
> Last, we introduce a confidence metric suitable with this pipeline, with operational thresholds for real-world use. We develop more on this topic in the answer to the second question.
>
> In summary, the novelty is not just in combining an autoencoder with
> VideoSeal, which is not possible, but in:
>
> -   **re-engineering and retraining a bit-based watermarker** into a
>     continuous, high-dimensional latent channel;
>
> -   **designing a robust text autoencoder** whose latent space is
>     compatible with this channel; and
>
> -   **integrating both with a security layer** to form an end-to-end
>     semantic watermarking framework;
>
> -  **derivating a confidence metric with operational threshold** to ensure trustworthiness at inference.

---

> ### Author Response · Authors · 2025-11-17
>
> > 2. Closeness between $\hat{y}$ and $\hat{y}_{rec}$ alone may not imply authenticity, as the relationship between $\hat{y}$ and the true latent vector $y$ remains unknown. Could the authors clarify how the confidence score in Equation (4) effectively captures this relationship?
>
> We agree for an in-distribution latent vector $y$, a well-trained
> autoencoder ideally satisfies $E(D(y)) \approx y$. However, *our setting
> crucially differs from this idealized case*: we do not know if the
> extracted latent $\hat{y}$ is in-distribution.
>
> First, $\hat{y}$ is produced by the extraction watermarking model
> applied to a potentially distorted image (JPEG, resize, blur, local
> edits, etc.). Even when the cover is not altered, the watermarker is not
> perfect: for clean images we obtain an average cosine similarity of
> about $0.987$ (cf Appendix Table 11) between the embedded and extracted
> vectors, which already departs from the identity case where
> $\hat{y}={y}$.
>
> Intuitively, $\hat{y}$ may lie in the correct region of the latent space but not exactly at the centroid of the corresponding message. Passing it through the autoencoder, $\hat{y}{\text{rec}} = E(D(\hat{y}))$, projects it to the nearest message centroid. Equation (4) then evaluates the probability of observing the empirical cosine similarity $\cos(\hat{y}, \hat{y}{\text{rec}})$ under the null hypothesis that the two vectors are independent. This yields a calibrated confidence score: small $p$-values indicate that $\hat{y}$ and $\hat{y}_{\text{rec}}$ are too close to be explained by chance, which we use at inference time to filter out untrustworthy messages and covers.
>
> Table 4 provides operational thresholds on this score to achieve target
> false positive rates, making the non-triviality of the proximity
> $\hat{y}$ and $\hat{y}_{rec}$ quantitatively explicit.
>
> This complement the previous answer, as this way of ensuring the trustworthiness of the received message is a new addition in comparison with VideoSeal.

---

> ### Author Response · Authors · 2025-11-17
>
> > 3. Secret rotation. What is the motivation for introducing the secret rotation layer? Please provide an example or real-world scenario where this component is necessary or advantageous for security or robustness.
>
> Thank you for this question. It addresses a key component of our
> framework's practical security.
>
> Our motivation follows a standard cryptographic principle: security
> should rely on a secret key rather than on the obscurity of the
> algorithm (Kerckhoffs' principle, Chapter 2 from \[1\]). The use of a secret key is a classic property of watermark system (Section 2.3.8 from \[1\]).
> The secret rotation layer is our mechanism for introducing such a key, providing **message confidentiality** at minimal cost.
>
> To illustrate a realistic use case, consider a content creator, Alice,
> who embeds provenance data into her images:
>
> 1.  **Message encoding.** Alice defines a message $m$ (e.g., authorship
>     and generation details), which is encoded into a latent vector
>     $y = E(m)$.
>
> 2.  **Keyed rotation (security step).** Using her secret key $k$, Alice
>     deterministically generates a rotation matrix
>     $\mathbf{O} \in \mathrm{SO}(256)$ (Appendix C) and computes the
>     secured latent $y_r = \mathbf{O} y$, which is then embedded into the
>     image.
>
> 3.  **Verification.**
>
>     -   **Unauthorized party (Eve).** An attacker with access to the
>         watermarked image and our public models can extract a rotated
>         vector $\hat{y}_r$, but without $k$ cannot reconstruct
>         $\mathbf{O}$ or its inverse. They therefore cannot recover $y$.
>         Feeding $\hat{y}_r$ directly to the decoder $D(\hat{y}_r)$
>         produces an unintelligible message, protecting the content.
>         Not using a secret key at all would expose the message to Eve and enable easy replacement, thus breaking trust in this framework.
>
>     -   **Authorized verifier (Bob).** A verifier with Alice's key $k$
>         reconstructs $\mathbf{O}$, computes
>         $\hat{y} = \mathbf{O}^\top \hat{y}_r$, and decodes $D(\hat{y})$
>         to recover $m$ and verify provenance.
>
> **Why this mechanism fits our framework.**
>
> -   It is applied **post-training**, so a single public model can
>     support arbitrarily many secret keys without retraining.
>
> -   Orthogonality makes the transform exactly invertible via a transpose
>     operation.
>
> -   It is **computationally cheap**, adding negligible overhead at
>     inference time (Appendix C).
>
> Finally, to directly address the point about robustness: this layer is
> used exclusively for **security (confidentiality)**. Robustness to image
> perturbations (e.g., JPEG, resize, blur) rely on the watermarking
> architecture and training procedure, and is essentially unaffected by
> the presence or absence of the rotation.
>
> \[1\] Cox I J, Miller M L, Bloom J A, et al. Digital watermarking\[J\].
> Morgan Kaufmann Publishers, 2008, 54: 56-59.
>
>
> > 4. Effect of latent dimensionality.
>
> We thank the reviewer for raising this point. In Appendix Table 7, we
> report a hyperparameter sweep over latent dimensionalities
> $\{128, 256, 384\}$ for the text autoencoder. Among these choices,
> $256$ dimensions achieve the best trade-off between reconstruction
> quality and robustness under the noise injection used during training.
> Since the autoencoder is the bottleneck of the overall system, we focus
> the watermarking experiments on this best-performing configuration.
>
> To summarize these results, we will add the following table to the main
> text for clarity.
>
> **Best robust text autoencoder configuration per latent dimension (Wikitext, 30 tokens)**
>
> | Latent dim $$z$$ | Hidden $$d$$ | Layers $$N$$ | LR      | Batch | LoRA $$r$$ | BLEU-4 ↑ | Loss ↓   |
> |:----------------:|:------------:|:------------:|:--------|:-----:|:----------:|:--------:|:--------:|
> | **128**          | 512          | 8            | 3.0e-4  | 256   | 8          | 43.62    | 0.4110   |
> | **256**          | 512          | 10           | 2.5e-4  | 128   | 32         | **98.72**| **0.0343** |
> | **384**          | 512          | 6            | 5.0e-4  | 256   | 32         | 82.22    | 0.1936   |
>
> In this sweep (Appendix Table 7), we explore many configurations and
> apply permissive early stopping when the validation BLEU-4 plateaus. For latent
> dimensions $128$ and $384$, training quickly stagnates and fails to
> match the performance of the $256$-dimensional model, which appears to
> strike a favorable balance between latent capacity and robustness to the
> training noise added to the latent. Keep in mind that this table report the best run for each latent dimension. So with the dimension 384, while not reported, 10-layers-config performed worse.
>
> We will make this result explicit in the main text (Section 4, training
> paragraph) and clarify that the watermarking component can in principle
> operate with other latent sizes, but we select $256$ as the empirically
> optimal one in our experiments.

---

> ### Author Response · Authors · 2025-11-17
>
> > 5. Missing comparisons: DwtDct, StableSignature, RAW.
>
> We thank the reviewer for this suggestion. We will explicitly mention
> DwtDct, StableSignature, and RAW in the related work section (DwtDct
> already appears in Appendix Table 6) and clarify their operating regime.
>
> DwtDct is a classical post-hoc multibit watermarking scheme with a
> payload of 32 bits at PSNR $\approx 40$ dB.
> StableSignature, on the other hand, is an *in-generation* watermark that
> modifies a specific generative model and is not applicable to arbitrary,
> already-generated images. It is designed to embed a short model
> signature (48 bits) into the samples of that particular generator, which
> is a fundamentally different use case from our post-hoc, model-agnostic
> setting. It is possible to perform post-hoc with it but only at the cost of encoding and decoding the image within its VAE, thus strongly degrading image quality, with PSNR $\approx 30$ dB.
>
> RAW is a zero-bit detector aimed at deciding whether a watermark is
> present in an image, rather than transmitting a payload. Reported AUC-ROC is aimed at watermark *detection*. In LatentSeal,
> the core question is not "is there a watermark?" but "can we reliably
> recover a message encoded as a latent vector?". Our metrics therefore
> quantify *message recovery* quality (and its calibrated confidence),
> which is not directly comparable to detection-oriented AUC-ROC as our setting is much more demanding. On a sidenote, the code repository does not support inference on images, only videos with frames of dim 512x512, while our framework is able to watermark images as little than 256x256.
>
> Due to these differences in setting (post-hoc vs in-generation),
> objective (detection vs message recovery), and payload regime (0--48
> bits vs continuous), running direct experimental baselines with
> StableSignature or RAW would not provide a meaningful comparison.
> Instead, in the revised version we will (i) clearly distinguish
> detection-oriented and message-recovery-oriented watermarking, and (ii)
> extend Appendix Table 6 with a short summary of these methods, their
> payload capacities, and their operating regime to give a complete
> landscape view.
>
> Here is the summary:
> | Watermarking method                         | Payload capacity (bits)                  | PSNR (dB) |
> |--------------------------------------------|------------------------------------------|-----------|
> | LatentSeal, (Ours)                          | >256 (continuous)                                     | 40–44     |
> | VideoSeal, Fernandez et al. (2024)          | 256                                      | 40–44     |
> | MBRS, Jia et al. (2021)                     | 64–625                                   | 36        |
> | TrustMark, Bui et al. (2023)                | 100                                      | 40–44     |
> | StegaStamp, Tancik et al. (2020)            | 32–64                                    | 42        |
> | CIN, Ma et al. (2022)                       | 30                                       | 36–39     |
> | WAM, Sander et al. (2025)                   | 32/region                                | 40        |
> | RivaGAN, Zhang et al. (2019)                | 32                                       | 40        |
> | DwtDct, Cox et al. (2008)                      | 32                                       | 40        |
> | StableSignature, Fernandez et al. (2023)    | 48 bits (model signature)                | 30        |
> | RAW, Xian et al. (2024)                     | 0 (presence detection only)              | 27        |

---

> ### Author Response · Authors · 2025-11-25
>
> Dear Reviewer,
>
> We wanted to follow up to ensure that our responses to your comments were satisfactory. Please feel free to reach out if you require any further clarifications. And thank you again for your time!

---

> ### Comment · Reviewer_AGKw · 2025-11-26
>
> I thank the authors for their rebuttal efforts. Nevertheless, several of my major concerns remain insufficiently addressed. In particular:
>
> 1. Novelty and relation to VideoSeal. The adaptation from embedding a binary sequence to a continuous vector appears to be an incremental extension, rather than a substantial theoretical or methodological advancement. Moreover, the EM metric is somewhat specific and limited in interpretability: it is zero in Table 4 because VideoSeal constrains the bit length to 256. This does not constitute a fundamental distinction, as (1) VideoSeal could, in principle, increase its embedding capacity (albeit with a potential trade-off in robustness), and (2) the proposed method performs worse than VideoSeal on other metrics such as B4 in most cases.
>
> 2. The response to my earlier question does not directly address my concern on the relationship between $\hat{y}$ and the true latent vector $y$. This relationship remains unknown when only $\hat{y}$ and  $\hat{y}_{rec}$ are available. The information about $y$ is zero given $\hat{y}$ and $\hat{y}{rec}$ solely. As a result, it is still unclear how authenticity is determined, specifically, one can verify that $\hat{y}$ is faithfully represents $y$. In addition, can you please clarify whether it is $y$ or $\hat{y}$ in Eq. (4), and how to prove this result if it is indeed $y$?
>
> 3. Comparison with related work. I would like to reiterate earlier point regarding comparisons with existing methods. While I understand that the objectives and procedures may differ, including such comparisons would meaningfully contextualize the contribution and help readers understand the trade-offs or additional costs associated with embedding semantically meaningful messages.

---

> > ### Author Response · Authors · 2025-11-27
> >
> > We thank the reviewer for their valuable feedback and time spent on our paper.
> >
> > Regarding the relationship between $y$ and $\hat{y}$: we appreciate you pressing on this point, as it helped us identify an error in Equation (4). The equation should reference $\hat{y}$, not $y$, the surrounding text does mention $\hat{y}$ correctly, but we missed updating the equation itself.
> >
> > To further clarify the intent: Equation (4) is not designed to verify the relationship between the extracted latent $\hat{y}$ and the ground-truth $y$. Rather, it measures whether the extracted $\hat{y}$ is self-consistent through the encode-decode cycle, allowing us to assess confidence in the decoded message. In other words, we are quantifying the trustworthiness of the recovered message, not attempting to prove correspondence to an unknown original. Table 4 shows operational points to use this score in real-world setting.
> >
> > We will correct this in the revised version. Thank you again for helping us catch this.

---

> > ### Author Response · Authors · 2025-11-27
> >
> > >Novelty and relation to VideoSeal. The adaptation from embedding a binary sequence to a continuous vector appears to be an incremental extension, rather than a substantial theoretical or methodological advancement. Moreover, the EM metric is somewhat specific and limited in interpretability: it is zero in Table 4 because VideoSeal constrains the bit length to 256. This does not constitute a fundamental distinction, as (1) VideoSeal could, in principle, increase its embedding capacity (albeit with a potential trade-off in robustness), and (2) the proposed method performs worse than VideoSeal on other metrics such as B4 in most cases.
> >
> > Thank you for this valuable feedback, which helps us clarify some important part of our work.
> >
> > We respectfully argue that LatentSeal represents a paradigm shift from multibit watermarking to continuous semantic optimization. Unlike standard baselines that treat payloads as fragile bit-strings, our approach optimizes cosine distance on the hypersphere $S^{255}$. This geometric formulation enables graceful degradation, where heavy attacks result in semantic drift rather than the catastrophic failure typical of arithmetic coding. Furthermore, this continuous representation supports specific architectural contributions: a fixed-size semantic bottleneck that avoids the truncation issues of variable-length LLM compression thanks to the autoencoder generative prior, a native key-conditioned rotation mechanism (SO(256)) for security, and a derived statistical confidence metric. These elements collectively overcome the rigidity and fragility limits of the current state-of-the-art bit-centric paradigm.
> >
> > The SBERT table below show LatentSeal graceful degradation (measured with SBERT) against VideoSeal+LLMZip on 1000 images with 256 bits messages from PixmoCap, declined on a set of 21 transforms. We manage to always achieve a better SBERT score, and under heavy attack (GaussianBlur, Jpeg50&40) we observe huge improvement.
> >
> >
> > | Transform   | LatentSeal | VideoSeal+LLMZip |
> > |-------------|------------|-----------|
> > | Identity    | 0.981      | 0.958     |
> > | HFlip       | 0.980      | 0.951     |
> > | Rot 5       | 0.961      | 0.904     |
> > | Rot 10      | 0.901      | 0.806     |
> > | Rot 90      | 0.961      | 0.921     |
> > | Crop 90     | 0.934      | 0.853     |
> > | GBlur 5     | **0.623**      | **0.305**     |
> > | Sat 0.5     | 0.981      | 0.957     |
> > | Sat 1.5     | 0.977      | 0.953     |
> > | Cont 0.5    | 0.975      | 0.948     |
> > | Cont 1.5    | 0.893      | 0.778     |
> > | Bright 0.5  | 0.970      | 0.939     |
> > | Bright 1.5  | 0.887      | 0.771     |
> > | Hue 0.1     | 0.948      | 0.859     |
> > | Hue -0.1    | 0.946      | 0.855     |
> > | Jpeg 80     | 0.923      | 0.888     |
> > | Jpeg 75     | 0.895      | 0.825     |
> > | Jpeg 70     | 0.865      | 0.758     |
> > | Jpeg 60     | 0.768      | 0.603     |
> > | Jpeg 50     | **0.659**      | **0.398**    |
> > | Jpeg 40     | **0.531**      | **0.193**     |
> >
> > We plan to add this table to the main text for better interpretability of our results.
> >
> > We think that all these contributions justify the need to explore further this research direction in watermarking.

---

> > ### Author Response · Authors · 2025-11-27
> >
> > 1.(1) While VideoSeal could theoretically increase its payload, watermarking methods face a hard "capacity-robustness-invisibility" triangle. VideoSeal is optimized to embed a bit-string robustly. Scaling this would require high perturbation magnitude that it would severely degrade image quality (invisibility) or robustness. Or, if allowed to be speculative, a relevant direction could be to just scale the architecture and x100 the number of parameters to achieve better results. But this would hamper the real-time capacity of the system, making it not suited for greater scale watermarking.
> > Our "fundamental distinction" is shifting from embedding arbitrary bit-strings to embedding continuous semantic latents. This allows us to bypass the standard capacity bottlenecks of watermarking by leveraging the generative prior of the autoencoder. And this is fundamental and totally unexplored in the watermarking literature.
> >
> >
> > (2) We deliberately included Wikitext as a stress-test for our method. This dataset is valuable because:
> >
> > 1. **High bit-length regime**: Wikitext samples are longer and more complex than typical image captions, requiring more bits to encode. This is the ">256 bits" regime where LLMZip must truncate, making it relevant for capacity comparisons.
> >
> > 2. **Transparency about limitations**: Wikitext's text with dense named entities represents a challenging scenario for our vocabulary coverage. We believe showing where our method struggles is important.
> >
> > Despite these challenges, LatentSeal achieves strong results: **51.6% exact match** with **median B4 = 1.0**, meaning the majority of samples are recovered perfectly, character for character. The reviewer raises a fair point about truncation affecting VideoSeal's EM score, but we view high EM as a capability our method provides that alternatives currently do not, valuable for applications requiring exact recovery such as authentication or legal watermarking.
> >
> > On mean B4, VideoSeal+LLMZip does achieve better scores. However, we believe this comparison is flawed due to two factors biasing it in their favor.
> >
> > **Wikitext is derived from Wikipedia**, which is dense with named entities, historical dates, and domain-specific terminology. This is precisely where our autoencoder struggles, not because of a fundamental limitation in the compression mechanism, but because these rare tokens are underrepresented in our training set.
> >
> >
> > Consider this abbreviated Wikitext test set sample:
> >
> > | Original | LatentSeal Output | VideoSeal Output|
> > |----------|-------------------|-----------------|
> > | Claudius Caesar | Caesarius Benedict | Claudius Caesar|
> > | Germanicus | 1577 | Germanicus|
> > | Roman Emperor | Soviet status | Roman Emperor|
> > | 15 December 37 AD | German December 1900 | 15 December 37 AD|
> >
> > The autoencoder learned *what* to encode (person + title + dates + role) and preserves the semantic structure, but lacks vocabulary coverage for rare historical tokens.
> >
> > Furthermore, as stated in Appendix D.1, **the baseline model (OPT-125m) was trained on Wikipedia**, which is the source corpus for Wikitext. Consequently, the test sequences are likely included in OPT's training data, which is used to perform LLMZip compression on the VideoSeal input for comparison.
> >
> > This means that for the LLMZip baseline, predicting "Claudius Caesar Augustus Germanicus" is easier because it is a likely memorized Wikipedia sequence. The same rare named entities that our autoencoder struggles to reconstruct are precisely those that OPT can retrieve from memory. This data leakage provides an unfair advantage to the baseline rather than reflecting a failure of our model to capture the content.
> >
> > Wikitext is a valuable stress-test that reveals our autoencoder's limitations with rare named entities, while also demonstrating the high bit-length regime where our capacity advantage matters. Despite being a challenging dataset, LatentSeal achieves 51.6% exact match with median BLEU-4 at 1.0.
> >
> > The direct comparison with VideoSeal+LLMZip on this dataset is biased against LatentSeal due to (1) the rare tokens disadvantaging our autoencoder, and (2) OPT's likely memorization of Wikipedia giving LLMZip an unfair advantage. This is meant as a stress test and we think it gives nuances to our work. We will make this stress test setting clearer and properly reference the Appendix about the test leak for Wikitext in the OPT model  within the main text.
> >
> > We thank you for the feedback, we think it will improve our submission.

---

> > ### Author Response · Authors · 2025-11-27
> >
> > >3. Comparison with related work. I would like to reiterate earlier point regarding comparisons with existing methods. While I understand that the objectives and procedures may differ, including such comparisons would meaningfully contextualize the contribution and help readers understand the trade-offs or additional costs associated with embedding semantically meaningful messages.
> >
> > We appreciate this suggestion. While a comprehensive comparison across all methods extends beyond our current scope, we propose a focused experiment that directly addresses your concern about trade-offs. We will provide ROC curves comparing: (1) existing methods using p-values from their respective papers, (2) LatentSeal with random watermarks ($y$ sampled uniformly from $\mathbb{S}^{255}$), and (3) LatentSeal with semantic watermarks ($y$ from our encoder). This will quantify any detection cost associated with embedding meaningful messages. Please let us know if this protocol aligns with your expectations.

---

> ### Author Response · Authors · 2025-11-28
>
> >3. Comparison with related work. I would like to reiterate earlier point regarding comparisons with existing methods. While I understand that the objectives and procedures may differ, including such comparisons would meaningfully contextualize the contribution and help readers understand the trade-offs or additional costs associated with embedding semantically meaningful messages.
>
>
> We agree that placing our method in the context of existing baselines is essential. To address this, we have computed detection performance (measured as $-\log_{10}(p)$ following standard procedures [1]) for LatentSeal against seven state-of-the-art watermarking methods. We also performed a specific ablation to isolate the "cost" of semantic embedding.
>
> To directly answer your question regarding the "additional costs associated with embedding semantically meaningful messages," we evaluated LatentSeal in two modes:
> 1. **Encoded:** where $y$ is the output of our semantic text autoencoder.
> 2. **Random:** where $y$ is sampled uniformly from $\mathbb{S}^{255}$ (simulating a zero-bit carrier).
>
> As shown in the "TPR at Extreme FPR" Table 2 below, the difference in detection power between these two modes is negligible. This indicates that **enforcing semantic structure incurs effectively zero cost** in terms of robustness or detectability compared to embedding random vectors.
>
> For further comparison against baselines, we report the detection strength as $-\log_{10}(\text{p-value})$. As shown in the Table 1 below, LatentSeal significantly outperforms all baselines, results reported from [1], including the state-of-the-art VideoSeal. We run transforms on 1000 watermarked images from COCOVal2017 with LatentSeal and VideoSeal to perform this study.
>
> This gap is structural: discrete multi-bit systems are theoretically bounded by their payload size (e.g., a 256-bit system like VideoSeal has a hard ceiling of $p \approx 2^{-256} \approx 10^{-77}$). In contrast, LatentSeal operates in a continuous latent space where the p-value is a function of cosine similarity, allowing it to provide statistical guarantees orders of magnitude stronger (exceeding $10^{-190}$).
>
> **Table 1: Detection Performance ($-\log_{10}(p)$)** (Higher is better)
>
> | Method | HiDDeN | MBRS | TrustMark | Stable Sig. | VideoSeal | **LatentSeal (Ours)** |
> | :--- | :--- | :--- | :--- | :--- | :--- | :--- |
> | **Identity** | 14.2 | 70.6 | 29.9 | 13.6 | 76.4 | **191** |
> | **Valuemetric** | 10.8 | 59.8 | 27.4 | 12.5 | 73.3 | **161** |
> | **Geometric** | 5.5 | 3.3 | 8.5 | 9.8 | 74.8 | **164** |
> | **Compression** | 14.2 | 69.9 | 29.7 | 7.5 | 72.5 | **132** |
> | **Combined** | 2.6 | 0.4 | 0.8 | 9.3 | 60.9 | **91** |
>
> To visualize this "hard ceiling" limitation of bit-based methods, we compare True Positive Rates (TPR) at extreme False Positive Rate (FPR) thresholds. VideoSeal’s reliability collapses once the FPR surpasses its bit-depth limit ($10^{-77}$), whereas LatentSeal maintains robust detection.
>
> **Table 2: TPR at Extreme FPR Thresholds (Averaged)**
>
> | FPR | VideoSeal | LatentSeal (Random) | LatentSeal (Encoded) |
> | :--- | :--- | :--- | :--- |
> | $10^{-50}$ | 94.3% | 96.3% | 96.2% |
> | $10^{-77}$ | 60.9% | 88.5% | 88.5% |
> | $10^{-80}$ | **0%** (Bit-depth Limit) | 87.6% | 87.5% |
> | $10^{-100}$ | **0%** | 79.2% | 79.2% |
>
> In summary, LatentSeal not only allows for the transmission of semantic messages but does so while providing **~2.5$\times$ stronger statistical guarantees** than the best available discrete baselines.
>
> We thank you for prompting this comparative analysis. It has allowed us to empirically demonstrate a critical finding: embedding semantic meaning incurs no robustness penalty compared to random vectors, they statistically improve detection. This reinforces our core argument that moving from discrete bit-spaces to continuous latent spaces offers superior performance without the traditional trade-offs. We hope these additional experiments fully address your concerns.
>
> We plan to add this comprehensive study in the revised version if you feel satisfied.
>
> We would like to thank you again for your valuable feedback.
>
> [1] Fernandez, P., Elsahar, H., Yalniz, I. Z., & Mourachko, A. (2024). Video Seal: Open and Efficient Video Watermarking. *arXiv preprint arXiv:2412.09492*.

---

### Official Review · Reviewer_zc4C · 2025-11-04

**Soundness:** 3
**Presentation:** 3
**Contribution:** 3
**Rating:** 8
**Confidence:** 2

**Summary:**

This paper proposes an image watermarking framework called LatentSeal that, instead of embedding arbitrary bits, embeds a 256D vector, which represents full-sentence textual messages (actually a set of tokens, e.g., 30). The mapping is performed with a text auto-encoder, where the encoder is derived from ModernBERT.
The latent vector is robustly embedded into the image using a finetuned watermarking model adapted from VideoSeal.
To ensure secrecy, the latent vector is secured through a secret, invertible rotation conditioned by a key, meaning only authorized decoders can correctly reverse the rotation and recover the message

The paper claims that the system is fast, secure, and offer a higher capacity than traditional bit-centric watermarking methods, while maintaining robustness against image attacks.

**Strengths:**

The problem related to data protection in the era of foundation models is quite important, and watermarking is certainly a key stream of research to that end.

The paper is well written and the contribution seems reasonable, to the best of my judgement (I have not followed closely the literature on this topic recently).

The lightweight design of the decoder should enable fast, real-time decoding, making it practical for real-world deployment

The code is actually provided in the supplemental material. 1

**Weaknesses:**

The paper explicitly states a major limitation: the robustness of the system against powerful, image-editing models is not addressed. The authors acknowledge that these models may be able to strip their watermark.

Minor: The font is two small in Figure 2.

**Questions:**

What happens if you encode random ids with your text auto-encoder?

---

> ### Author Response · Authors · 2025-11-14
>
> We sincerely thank the reviewer for the positive assessment, the clear
> summary of our contributions, and the encouraging rating.
>
> ### On the strengths highlighted by the reviewer.
>
> We are grateful that the reviewer emphasizes: (i) the importance of data
> protection and watermarking in the era of foundation models, (ii) the
> clarity of the presentation, (iii) the practicality of the lightweight
> decoder, and (iv) the availability of code.
>
> We fully agree that practicality is crucial for real-world deployment.
> In this regard, we stress that the decoder was explicitly designed to be
> lightweight and fast: as shown in Fig. 3, our decoder achieves up to
> $121\times$ higher decoding throughput than the Videoseal+LLMZip
> baseline at batch size 128, while maintaining robustness to common image
> transformations. We believe this aligns with the reviewer's point that
> the framework is suitable for real-time or large-scale use.
>
> We thanks the reviewer's for noting the code is available, and want to add that **the models will also be released**. We think open-sourcing is beneficial for the community. The modular design of LatentSeal (separate text autoencoder, watermarker, and security layer) is intended to make our system easy to
> adopt and extend, and the provided code is structured so that each
> component can be reused independently (e.g., the latent watermarker for
> non-text payloads, the autoencoder for semantic compression).
>
> > What happens if you encode random IDs with your text autoencoder?
>
> It fails, as our text autoencoder takes token sequences as input, using a
> ModernBERT-based encoder and a learned decoder. It is trained on
> natural-language sentences, so its latent space is shaped to represent
> *semantic text*, not arbitrary identifiers. Sampling randomly token ids break the assumption that the message follows training set distribution, representative of natural-language rules.
>
> To verify this empirically, we conducted a simple experiment: we
> sampled 30 random token ids from ModernBert vocabulary, this constitues a random sentence. We pass through the autoencoder (encode $\rightarrow$ decode), and we compute
> exact string reconstruction and BLEU4.
>
> Over $1000$ such sentences, the exact-match rate was **0 %** and the average BLEU4 is of 0.0029: *none*
> of the decoded outputs matched the input sentence. Instead, the decoder
> systematically produced linguistically plausible fragments that lie on
> the natural-language manifold, confirming that the model projects
> such out-of-distribution inputs toward nearby semantic regions in the
> latent space rather than memorizing arbitrary identifiers.
>
> To illustrate this behavior qualitatively, consider the following
> example from our random-token-ids experiment:
>
> > **Input (30-token random ids):**\
> > raw_token_ids=[43008, 23303, 36717, 12080, 15091, 24052, 44294, 28853, 36347, 418, 13265, 1339, 33572, 5884, 30018, 39866, 15097, 45563, 18594, 39954, 12753, 39028, 4893, 7938, 5418, 30947, 31703, 27902, 13990, 44894]
> >
>   >`text input:  '083ovich064pay lawsuit hypers phenomenalshots Resolution L twelve let Tigers minor sadly fussÙstrengthоп upholdNumdq applications faster Black NovaSerializerSync eleg diseng`\
> >
> > **Decoded output (autoencoder, no watermarking):**\
> > `![AIwor pandemic transactionancellies Impact confirmation No Notice " Midlands Male SimplyDesign efficiently ruthatieallowedBindingacity Plansrives standard Beluba`
>
> We plan to add this experiment to the Appendix to illustrate this specific use case which is not suited for our method.
>
> We hope this answer your question.

---

### Author Response · Authors · 2025-11-18
**Summary of the discussion and subsequent revisions for the AC**

We sincerely thank the reviewers for their highly constructive feedback. The primary concerns raised centered on the paper's novelty, the lack of statistical comparison with more baselines for the detection task, and the justification for our continuous-latent design, with additional concerns about imperceptibility and security.

The revised manuscript incorporates substantial changes, including new experiments and several correction, that definitively address these points.

## 1. Core novelty and paradigm shift (Reviewers AGKw, era7)

The core criticism, that LatentSeal is an incremental extension of VideoSeal, is addressed by clarifying that our work constitutes a **paradigm shift** from discrete to continuous semantic watermarking.

| Reviewer Concern | Resolution & New Evidence | Impact |
| :--- | :--- | :--- |
| Novelty is low; it's just "Autoencoder + VideoSeal." | **Reframed Core Innovation:** We clarified that the work requires re-engineering the watermarker from a **discrete bit-pipe** into a **continuous vector channel** ($\mathbb{S}^{255}$), demanding architectural changes and a new **cosine-similarity loss**. This continuous space is the foundation for all subsequent benefits and state-of-the-art results. | **Section 4 & Appendix B** emphasize that the novelty lies in the **channel adaptation** and the resulting properties. |
| Claim of "breaking the 256-bit limit" is unclear. | **Clarified Practical Scope:** The **Abstract** was revised to explicitly state that the 256-bit ceiling is an *empirical observation* from baselines, framing our achievement as a **practical, baseline-relative breakthrough** in capacity. | **Abstract** clarifies the scope, and **Table 4** (and Appendix D.3) shows reliable recovery in the "high payload" regime where baselines must truncate. |
| Continuous vectors offer no benefit. | **New SBERT Evidence:** We added Table 14 showing LatentSeal achieves **significantly better graceful semantic degradation** under heavy attacks (e.g., JPEG 40, Gaussian Blur) compared to the catastrophic failure typical of discrete methods like VideoSeal. | **Table 14** now empirically proves the value of the continuous representation for semantic robustness while **Section B.3** elaborates on statistical guarantees at detection. |

## 2. Confidence metric and detection setting (Reviewer AGKw)

We corrected a typo error in Equation (4) and added teh requested comparison for the detection task, introducing theoretical settings for a reliable comparison.

| Reviewer Concern | Resolution & New Evidence | Impact |
| :--- | :--- | :--- |
| Confidence score (Eq. 4) is theoretically flawed. | **Correction of Typo & Clarification of Intent:** We corrected a critical typo in **Equation (4)** (changing y to $\hat{y}$), resolving the ambiguity. We clarified that the metric is a **self-consistency check** designed to determine the **trustworthiness of the *recovered* message**. | **Equation (4)** is now formally sound, and its role as a calibrated score is justified via **Table 5 (Operational Thresholds)**. |
| Need for rigorous comparison using p-values. | **New p-value Comparative Study:** We performed a comprehensive study and added **Table 12** (Detection Performance) and **Table 13** (Extreme FPRs). This evidence shows LatentSeal provides **statistical guarantees orders of magnitude stronger** than discrete systems, which hit a hard ceiling at their bit-depth (~$10^{-77}$). | **Appendix B.3** now provides the theoretical analysis demonstrating LatentSeal's superior statistical guarantees with respect to previous baselines. |
| Is there a detection cost for embedding meaningful vs. random vectors? | **Specific Ablation Study:** We added a direct ablation in **Table 13** comparing autoencoder-encoded vs. random vectors. The results show no significant difference, empirically proving that **enforcing semantic structure incurs zero penalty**. | **Table 13** empirically answer the trade-off concern. |

---

> ### Author Response · Authors · 2025-12-01
> **Summary of the discussion and subsequent revisions for the AC - Part 2**
>
> ## 3. Justification of Design and Context (Reviewers AGKw, era7)
>
> We clarified the motivations behind key implementation choices (training, security layer) and provided missing context.
>
> | Reviewer Concern | Resolution & New Evidence | Impact |
> | :--- | :--- | :--- |
> | Why not end-to-end  training? | **Justified Modularity:** **Section 4** was expanded to explain that modularity was chosen to: (1) allow the watermarker to generalize to **any continuous latent** (cross-modal use), (2) avoid the significant optimization complexity of coupled end-to-end training, and (3) preserve the use of the **key-conditioned rotation** security layer. | **Section 4** provides a clear, principled defense of the training strategy. |
> | Motivation for secret rotation layer? When is it useful ? | **Provided practical use case:** **Section 3.3** was expanded to provide a concrete **provenance verification scenario** and justify the layer based on **Kerckhoffs' principle**, using a cheap, post-hoc key to ensure **message confidentiality**. | **Section 3.3** clarifies that this is a *security* mechanism, not a robustness mechanism and advocates for the use of secret key in watermarking. |
> | Missing watermark strength/imperceptibility metrics. | **New Perceptibility Metrics:** We highlighted that all main experiments were run at 42 dB PSNR. We added a new **Table 3** showing the relationship between the scaling factor $\alpha$ and multiple perceptibility metrics (**PSNR, SSIM, LPIPS**) and cosine similarity, fully characterizing the trade-off. | **Section 4.1** now provides comprehensive metrics to control and verify imperceptibility. |
> | Need for full baseline context. | **Updated Baselines and Context:** **Table 16 (Appendix)** was updated to include DwtDct, StableSignature, and RAW, explicitly categorizing them by their **payload limits** (e.g., 32-48 bits) and objective (detection vs. message recovery), completing the landscape view. | **Appendix Table 16** fully addresses the contextualization request. |

---

### Meta-Review · Area_Chair_NgwU · 2026-01-05

**Summary:**

This paper proposed LatentSeal that reframed watermarking as semantic communication: a lightweight text autoencoder maps full-sentence messages into a compact 256-dimensional unit-norm latent vector, that embedded by a finetuned watermark model. The method achieves robust and high-capacity watermarking.

this paper got one 8 rating,  one 2 rating and one 4 rating.

The strength of this paper given by reviewers are:
1. the problem is quite important. (Reviewer zc4C)
2. paper is well written and contribution seems reasonable. (Reviewer zc4C, AGKw)
3. practical for real-world deployment. (Reviewer zc4C)
4. code is provided. (Reviewer zc4C)
5. idea is appealing. (Reviewer AGKw, era7)
6.  effectiveness and robustness is verified. (Reviewer era7)

The weakness of this paper given by reviewers are:
1. the robustness of the system against powerful, image-editing models is not addressed. (Reviewer zc4C)
2. Comparison to existing work. (Reviewer AGKw)
3. The detection mechanism and its justification remain unclear. (Reviewer AGKw)
4. motivation for introducing the secret rotation layer? (Reviewer AGKw)
5. generality. (Reviewer AGKw)
6. add more baseline methods. (Reviewer AGKw)
7. structure of the proposed model is relatively simple and seems to be a combination of existing works. (Reviewer era7)
8. The experimental metrics only consider BLEU-4 and EM, without incorporating measures related to watermark strength. (Reviewer era7)

Questions:
1. What happens if you encode random ids with your text auto-encoder? (Reviewer zc4C)
2. Where does the core innovation of this paper lie? (Reviewer era7)
3. Could an end-to-end training approach be considered instead? (Reviewer era7)
4. The paper claims to "break the 256-bit payload limit," but this is achieved by compressing the text into a continuous vector space, which seems to differ from the traditional definition of "bits" in information theory. Could you clarify this? (Reviewer era7)

Reviewer AGKw and era7 engaged in the discussion and suggested their concerns are not addressed. Given these AC decision to reject this paper. But hope authors will found reviewers' comments are useful for their future research.

**Reviewer Concerns:**

weakness 1. authors didn't provide answer to this.

weakness 2. authors mentioned they operate at a different operational domain, using different learning objective and the watermark could embed high-dimensional, continuous unit-norm vectors into images.

weakness 3. authors provide more detailed explanations.

weakness 4. authors provided more detailed explanations.

weakness 5. authors mentioned they have results in appendix table 7 shows 256 achieve the best results.

weakness 6. no real comparison provided.

weakness 7. not answered.

weakness 8. more results added.

**Reviewer Scores:**

Reviewer zc4C will probably keep their rating 8.

Reviewer AGKw mentioned their major concerns is not addressed and so they probably will keep rating 2.

Reviewer era7 mentioned they will keep rating 4.

---

### Decision · Program_Chairs · 2026-01-26

Reject